# Disentanglement with Biological Constraints: A Theory of Functional Cell Types

**James C.R. Whittington**[*]
Stanford & Oxford

**William Dorrell**
UCL

**Surya Ganguli**
Stanford

**Timothy E.J. Behrens**
UCL & Oxford

## Abstract

Neurons in the brain are often finely tuned for specific task variables. More-over, such disentangled representations are highly sought after in machine learning. Here we mathematically prove that simple biological constraints on neurons, namely nonnegativity and energy efficiency in both activity and weights, promote such sought after disentangled representations by enforcing neurons to become selective for single factors of task variation. We demonstrate these constraints lead to disentanglement in a variety of tasks and architectures, including variational autoencoders. We also use this theory to explain why the brain partitions its cells into distinct cell types such as grid and object-vector cells, and also explain when the brain instead entangles representations in response to entangled task factors. Overall, this work provides a mathematical understanding of why single neurons in the brain often represent single human-interpretable factors, and steps towards an understanding task structure shapes the structure of brain representation.

## 1 Introduction

Understanding why and how neurons behave is now foundational for both machine learning and neuroscience. Such understanding can lead to better, more interpretable artificial neural networks, as well as provide insights into how biological networks mediate cognition. A key to both these pursuits lies in understanding how neurons can best structure their firing patterns to solve tasks.

Neuroscientists have some understanding of how task demands affect both early single neuron responses (Olshausen & Field, 1996; Yamins et al., 2014; Ocko et al., 2018; McIntosh et al., 2016) and population level measures such as dimensionality (Gao et al., 2017; Stringer et al., 2019). However, there is little understanding of neural population structure in higher brain areas. As an example, we do not even understand why many different bespoke cellular responses exist for physical space, such as grid cells (Hafting et al., 2005), object-vector cells (Høydal et al., 2019), border vector cells (Solstad et al., 2008; Lever et al., 2009), band cells (Krupic et al., 2012), or many other cells (O'Keefe & Dostrovsky, 1971; Gauthier & Tank, 2018; Sarel et al., 2017; Deshmukh & Knierim, 2013). Each cell has a well defined, specific cellular response pattern to space, objects, *or* borders, as opposed to a mixed response to space, objects, *and* borders. Similarly, we don't understand why neurons in inferior temporal cortex are aligned to axes of data generative factors (Chang & Tsao, 2017; Bao et al., 2020; Higgins et al., 2021), why visual cortical neurons are de-correlated (Ecker et al., 2010), why neurons in parietal cortex are selective only for specific tasks (Lee et al., 2022), why prefrontal neurons are apparently mixed-selective (Rigotti et al., 2013), and why grid cells sometimes warp towards rewarded locations (Boccara et al., 2019) and sometimes don't (Butler et al., 2019). In essence, why are some neural representations entangled and others not?

Machine learning has long endeavoured to build models that disentangle factors of variation (Hinton et al., 2011; Higgins et al., 2017a; Locatello et al., 2019; Bengio et al., 2012). We define disentanglement as single neurons responding to single factors of variation (see Appendix A.1 for further details). Such disentangled factors can facilitate compositional generalisation and reasoning (Higgins et al., 2018; 2017b; Whittington et al., 2021a) (though some work has challenged the idea that disentangled representations generalise better (Schott et al., 2022), as well as lead to more interpretable outcomes in which individual neurons represent meaningful quantities. Unfortunately, building models that disentangle is challenging (Locatello et al., 2019).

---

[*]Correspondence to: `jcrwhittington@gmail.com`

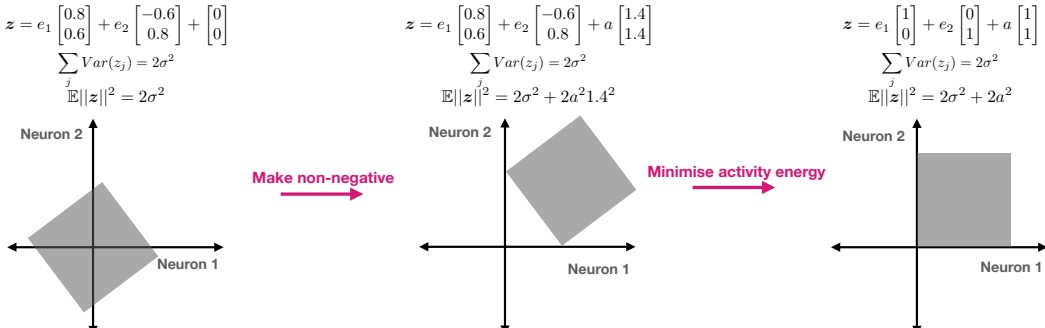

Figure 1: **Proof intuition.** Two uniformly distributed independent factors represented with two entangled neurons (left). The representation can be made nonnegative at the expense of activity energy (middle). Activity energy is minimised under a nonnegativity (and variance) constraint when the neurons are axis aligned to task factors (i.e. disentangled, right). Grey boxes denote uniform distributions over neural activity induced by uniform distributions over task factors. Note our proof does not require uniformity.

In this work we 1) prove simple biological constraints of **nonnegativity** and **minimising activity energy** lead to factorised representations in linear networks; 2) empirically show these constraints lead to disentangled representations in both linear and nonlinear networks; 3) obtain competitive disentanglement scores on a standard disentanglement benchmark; 4) provide an understanding why neurons in the brain are characterised into specific cell types due to these same biological constraints; 5) empirically show these constraints lead to specific cell types; 6) suggest when and why neurons in the brain exhibit disentanglement versus mixed-selectivity.

Please see appendix A.2 for a comprehensive discussion relating our work to existing literature.

## 2 LINEAR DISENTANGLEMENT WITH BIOLOGICAL CONSTRAINTS

We first provide a theorem that suggests why the combined biological constraints of nonnegativity and energy efficiency lead to neural disentanglement (proofs of all theorems are in App. A.3):

**Theorem 1**. Let $e \in \mathbb{R}^k$ be a random vector whose $k$ independent components denote $k$ task factors. We assume each independent task factor $e_i$ is drawn from a distribution [1] that has mean 0, variance $\sigma^2$, and maximum and minimum values of $\min(e_i) = -a$ and $\max(e_i) = a$. Also let $z \in \mathbb{R}^n$ be a linear neural representation of the task factors given by

$$z = Me + b_z, \tag{1}$$

where $M \in \mathbb{R}^{n \times k}$ are mixing weights and $b_z \in \mathbb{R}^n$ is a bias. We further assume two constraints: (1) the neural representation is *nonnegative* with $z_i \geq 0$ for all $i = 1, \ldots, n$, and (2) the neural population variance is a nonzero constant, $\sum_j Var(z_j) = C$, so that the neural representation retains some information about the task variables. Under these two constraints we show that in the space of all possible neural representations (parameterised by $M$ and $b_z$), the representations that achieve minimal activity energy $\mathbb{E}||z||^2$ also exhibit disentanglement, by which we mean every neuron $z_j$ is selective for at most one task parameter: i.e. $|M_{jk}||M_{jl}| = 0$ for $k \neq l$, a.k.a. each row of $M$ has at most 1 non-zero entry. (Proof in App. A.3.1).

**Intuition.** The intuition underlying the proof of the theorem is shown in Fig.1, where the key idea can be seen with two neurons encoding two factors. In particular, the bias must make every $z_i$ nonnegative for all values of $e_1$ and $e_2$. But since $e_1$ and $e_2$ are independent, the minimum firing of neuron 1 for example obeys $\min(z_1) = \min(0.8e_1 - 0.6e_2) = \min(0.8e_1) + \min(-0.6e_2) = -a(0.8 + 0.6)$. Thus for neurons that mix factors, a larger bias term must be used to ensure non-negativity, which leads to increased expected energy. Minimising this energy (subject to a constant variance) requires the smallest possible bias for each neuron, which occurs when each neuron is selective for a single task factor. We note that this is consistent with neurons having a baseline firing

---

[1]We understand that task factors, or indeed neurons in the brain, are generally not i.i.d., but we have made this assumption for mathematical convenience.

rate where now activity below baseline corresponds to negative/positive values in the distribution, and activity above baseline corresponds to positive/negative values in the distribution.

The above theorem, while simple, is restricted in two ways: (1) the independent task factors $e_i$ are *directly* available to the network; (2) representational collapse (i.e. setting $z = 0$) under energy minimisation is prevented solely by a variance constraint. We thus consider a more general setting where a neural circuit receives not the independent task factor vector $e$, but instead receives the mixed combination $x = De$, where $x \in \mathbb{R}^m$, $D \in \mathbb{R}^{m \times k}$ and $m \geq n \geq k$. We further model the neural representation $z$ as a linear generative model that can predict observed data $x$ via $x = Wz + b_x$. Thus prediction, not variance constraint, now prevents collapsing neural representations (proof in Appendix A.3.2). Furthermore in Appendix A.3.3 we prove the following:

**Theorem 2.** Let $x = De$ be observed entangled data, where the independent task factor vector $e$ obeys the same distributional assumptions as in Theorem 1. Let a neural representation $z$ exactly predict observed data via $x = Wz + b_x$ with zero error. Then for all such data generation models (with parameters $D$) and all such neural representations (with parameters $W$ and $b_x$), as long as: (1) the columns of $D$ are (scaled) orthonormal; (2) the norm of the read-out weights $||W||_F^2$ is finite; (3) the neural representation is nonnegative (i.e. $z > 0$), then out of all such neural representations, the minimum energy representations are also disentangled ones. By this we mean that each neuron $z_i$ will be selective for at most one hidden task factor $e_j$.

We note that $D$ having (scaled) orthonormal columns may seem like a strong constraint, but it holds approximately for any random matrix $D$ with many observations (dimensionality of $x$) and few independent task factors (dimensionality of $e$) (proof in Appendix A.3.4). Appendix A.3.5 discusses and provides intuition for when disentanglement occurs as $D$ takes more general forms.

Strikingly, the essential content of Theorem 2 is that any linear, nonnegative, optimally energetically efficient, generative neural representation that accurately predicts entangled observations that are linear mixtures of hidden task factors, will possess single neurons that are selective for individual task factors, despite *never* having *direct* access to them. In terms of applications, this theorem could apply in supervised or self-supervised machine learning settings in any neural network layer $z$ that is linearly read-out from, or in neuroscience settings where $z$ reflects a neural population from which one attempts to linearly decode task factors. While Theorem 2 holds in a simple setting, we will show through simulations that its essential content, namely that nonnegativity and energy efficiency together promote disentanglement, holds in practice in much more complex multilayer neural networks.

## 3   DISENTANGLEMENT IN MACHINES

We now present simulation results demonstrating that nonnegativity and energy efficiency (minimising either activity or weight energy) lead to single neuron selectivity for single task factors. We show this for supervised and unsupervised learning, both for linear and nonlinear tasks and networks (details of datasets, models, and simulations in Appendix A.4 & A.5).

**A measure for disentangled subspaces.** While our theory describes when single neurons become selective for single independent task factors, it does not limit the number of neurons selective for any given factor. For example in Theorem 2, four copies of the same neuron in $z$, each with half the activity, along with four copies of projecting weights each with half the values, predicts $x$ just as well and has exactly the same energy in both $z$ and $W$ [2]. More interestingly, an underlying task factor may not be one-dimensional, e.g. spatial location, in which case the subspace that codes for this factor have at least the same dimension. This phenomena cannot be captured by many metrics of disentanglement (e.g. the popular mutual information gap; MIG; Chen et al. (2018)) since they score highly if each factor is represented in just *one* neuron. Thus we define a new metric (mutual information ratio; MIR) that instead scores highly if each neuron only cares about one factor (see Appendix A.1 for details).

**Regularizers as constraints.** We impose nonnegativity via a ReLU activation function, or softly via explicit regularization $\mathcal{L}_{\text{nonneg}} = \beta_{\text{nonneg}} \sum_i \max(-a_i, 0)$ where $i$ indexes a neuron in the network, and $\beta_{nonneg}$ determines the regularization strength. Similarly, we apply regularization to the activity

---

[2]Learning dynamics may favour fewer neurons per factor as there are fewer weights to align.

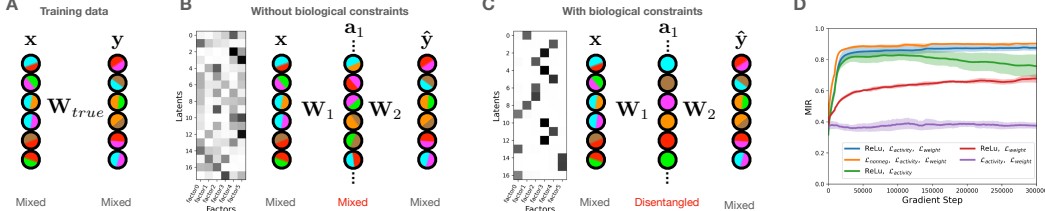

Figure 2: **Shallow linear networks disentangle.** We train 1-hidden layer linear networks on linear data. **A)** Cartoon schematic showing both input and output are entangled linear mixtures of factors (colours). Neurons colours schematically denote which of the factors it codes for. $\boldsymbol{W}_{true}$ is the true mapping between $\boldsymbol{x}$ and $\boldsymbol{y}$. **B)** A model without biological constraints learns entangled internal representations. Mutual information matrix (scale 0 to 0.6) shown on left, cartoon schematic on right. $\boldsymbol{W}_1$ and $\boldsymbol{W}_2$ are the learnable weights projecting to and from the hidden layer. **C)** A model with our constraints learns disentangled representations (MI matrix scale 0 to 2.25). **D)** Several model variants, in which only those with all our constraints learn disentangled representations (definition of metric MIR in Appendix A.1). Average and standard error shown for 5 random seeds.

energy and weight energy; $\mathcal{L}_{\text{activity}} = \beta_{\text{activity}} \sum_l ||\boldsymbol{a}_l||^2$ and $\mathcal{L}_{\text{weight}} = \beta_{\text{weight}} \sum_l ||\boldsymbol{W}_l||^2$. The role of $\mathcal{L}_{\text{weight}}$ is to promote activity (variance) in the network, otherwise activity could be reduced via $\mathcal{L}_{\text{activity}}$, and such reduced activity could be compensated for with arbitrarily large weights. The total loss we optimise is

$$\mathcal{L} = \underbrace{\mathcal{L}_{\text{nonneg}} + \mathcal{L}_{\text{activity}} + \mathcal{L}_{\text{weight}}}_{\text{Biological constraints}} + \underbrace{\mathcal{L}_{\text{prediction}}}_{\text{Functional constraints}} . \tag{2}$$

Here 'functional constraints', are any prediction losses the network has i.e. error in predicting target labels in supervised learning, or reconstruction error in autoencoders.

**Disentanglement in supervised shallow neural networks**. First we consider a dataset $\mathcal{D} = \{\boldsymbol{x}, \boldsymbol{y}\}$, where $\boldsymbol{x}$ is a orthogonal mixture of six i.i.d. random variables (hidden independent task factors; uniform distribution), and $\boldsymbol{y}$ is a linear transform of $\boldsymbol{x}$ (dimension 6; Fig. 2A). First we train shallow linear networks to read-out $\boldsymbol{y}$ from $\boldsymbol{x}$. Networks without biological constraints exhibit mixed internal representations (Fig. 2B). However, with our constraints, networks learn distinct sub-networks for each task factor (Fig. 2C). Removing any one of our constraints leads to entangled representations (Fig. 2D). Lastly we note sparsity constraints do not induce disentanglement (Appendix A.6). Thus the disentanglement effect of ReLUs is not from sparsity, but instead from nonnegativity.

**Disentanglement in supervised deep neural networks**. Training deep nonlinear (ReLU) networks on this data also leads to distinct sub-networks, with all layers learning disentangled representations (Fig. 3A). However with nonlinear data ($\boldsymbol{x} \leftarrow \boldsymbol{x}^3$, $\boldsymbol{y}$ remaining the same), the early layers are mixed-selective, whereas the later layers are disentangled (Fig. 3B-C). Understanding why the final hidden layer disentangles is easy, since it linearly projects to the target and so our theory directly applies. By extrapolating our theory, we conjecture that our biological constraints encourage any layer to be as linearly related to task factors and as disentangled as possible. However, early layers cannot be linear in hidden task factors since they are required to perform nonlinear computations on the nonlinear data, and thus only once activity becomes linearly related to independent task factors in later layers does disentanglement set in (as predicted by our linear theory).

**Disentanglement in unsupervised neural networks**. We now consider unsupervised learning, i.e. $\mathcal{D} = \{\boldsymbol{x}\}$, where $\boldsymbol{x}$ is a linear mixture of multiple independent task factors as in Theorem 2. Training 0-hidden layer autoencoders on this data, with our biological constraints, recovers the independent task factors in individual neural subspaces (Fig. 4A/B). Moreover, this only occurs when all constraints are present (Fig. 4A). Again, even though our theory applies to the linear setting, the same phenomena occur when training deep nonlinear autoencoders on nonlinear data; i.e. when $\boldsymbol{x}$ is a *nonlinear* mixture of multiple i.i.d. random variables, i.e. $\mathcal{D} = \{f(\boldsymbol{x})\}$ (Fig. 4C-D).

**Disentanglement on a standard benchmark with VAEs**. We now consider a standard disentanglement dataset (Fig. 5A; Kim & Mnih (2018). To be consistent with, and to compare to, the disentanglement literature we use a VAE and measure disentanglement with the familiar mutual-information gap (MIG) metric (Chen et al., 2018). For nonnegativity we ask the mean of the posterior to be

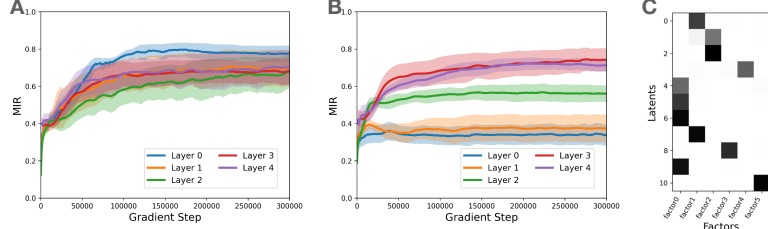

Figure 3: **Deep nonlinear networks disentangle.** We train 5-hidden layer nonlinear networks with our constraints on linear and nonlinear data. **A)** For *linear* data, all layers in the network learn a disentangled representation. **B)** For *nonlinear* data, only later layers learn a disentangled representations. **C)** Example mutual information matrix from the penultimate hidden layer.

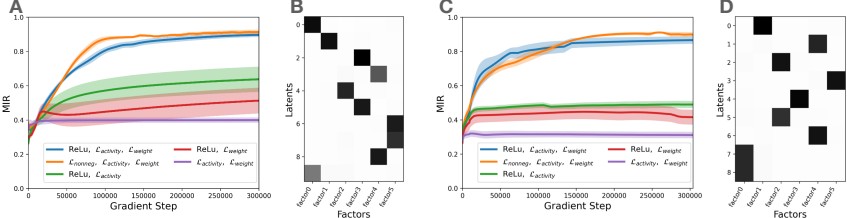

Figure 4: **Learning data generative factors with autoencoders. A)** Training linear autoencoders on linear data. Only models with our constraints learn disentangled representations. **B)** Example mutual information matrix from a high MIR model. **C)** Nonlinear autoencoders trained on nonlinear data. Only models with our constraints learn disentangled representations. **D)** Example mutual information matrix from a high MIR model. All learning curves show mean and standard error from 4 mean from 5 random seeds.

nonnegative (via a ReLU) [3], but we *do not* add a norm constraint as the VAE loss already includes one in its KL term between the Gaussian posterior and the Gaussian prior.

While state-of-the-art results are not our aim (instead we wish to elucidate that simple biological constraints lead to disentanglement), our disentanglement results (Fig. 5B) are competitive and often better than those in the literature (comparing to results of many models shown in Locatello et al. (2019)) even though those models explicitly ask for a factorised aggregate posterior. The particular baseline model we show here is $\beta$-VAE (Fig. 5D). We see that (1) our constraints lead to disentanglement (Fig. 5B-C); (2) including a ReLU improves $\beta$-VAE disentanglement (as predicted by nonnegativity arguments above, Fig. 5D); and (3) our constraints give results in the Goldilocks region of high disentanglement and high reconstruction accuracy (Fig. 5E).

## 4    DISENTANGLEMENT IN BRAINS: A THEORY OF CELL TYPES

We next turn our attention to neuroscience, which is indeed the inspiration for our biological constraints. While we hope our general theory of neural representations will be useful for explaining representations across tasks and brain areas, for reasons stated below, we choose our first example from spatial processing in the hippocampal formation. We show our biological constraints lead to separate neural populations (modules) coding for separate task variables, but **only** when task variables correspond to independent factors of variation. Importantly, the modules consist of distinct functional cell types with similar firing properties, resembling grid (Hafting et al., 2005) and object-vector cells (Høydal et al., 2019) (GCs and OVCs).

We choose to focus on spatial representations for two reasons. Firstly, there is a significant puzzle about why neurons deep in the brain, synaptically far from the sensorimotor periphery, almost miraculously develop single cell representations for human-interpretable factors (e.g. GCs for location in space; Fig. 6A, and OVCs for relative location to objects Fig. 6B). Such observations are not easily accounted for by standard neural network accounts that argue that representations are unlikely to be human-interpretable (Richards et al., 2019). Secondly, whilst these bespoke spatial

---

[3]Using a nonnegative posterior mean is odd when the prior is Gaussian, but it allows for easier comparison.

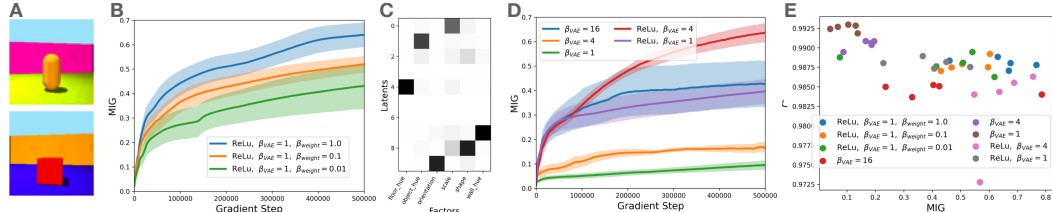

Figure 5: **Learning data generative factors with variational autoencoders. A)** We train on the Shapes3D dataset, with two example images shown. These images have 6 underlying factors. **B)** MIG scores are higher with higher weight regularization, and generally higher than any $\beta$-VAE (panel D). $\beta_{\text{weight}}$ is the regularisation strength of the weight regularisation . **C)** Mutual information matrix for a high scoring model. **D)** $\beta$-VAE MIG scores. Adding a ReLU improves MIG scores. **E)** MIG score against $R^2$ shows models with our constraints lie in the Goldilocks region of high disentanglement and high reconstruction. All learning curves show mean and standard error from 5 random seeds. Results from an additional dataset in Fig. 13

representations are commonly observed to factorise into single cells, there are situations in which selectivity spans across multiple task variables (Boccara et al., 2019; Hardcastle et al., 2017). For example, sometimes spatial firing patterns of GCs are warped by reward (Boccara et al., 2019) and sometimes they are not (Butler et al., 2019). There is no theory for explaining why and when this happens.

**A factorised task for rodents.** We consider a task in which rodents must know where they are in space, but must also approach one of multiple objects. If objects appear in different places in different contexts, the task is factorised into independent factors (Fig. 6C): 'Where am I in allocentric spatial coordinates?' and 'Where am I in object-centric coordinates?'. By contrast, if objects always appear in the same locations, the task is not factorised (as spatial location can predict object location). Formally, our task requires predicting spatial location, $\boldsymbol{x}$, whether an object is observed, $\boldsymbol{o}$, and the optimal action, $\boldsymbol{a}$. If objects move between tasks, then $p(\boldsymbol{x}, \boldsymbol{a}, \boldsymbol{o}) = p(\boldsymbol{x})p(\boldsymbol{a}, \boldsymbol{o})$, where $\boldsymbol{o}$ and $\boldsymbol{a}$ are not factored since optimal actions are dependent on objects (see Appendix A.9 for details). Our theory says the representation will have two subspaces - one for allocentric location (for predicting $\boldsymbol{x}$) and one for location relative to objects (for predicting $\boldsymbol{o}$ and $\boldsymbol{a}$) - and that these sub-spaces should be represented in separate neural populations when biological constraints are present.

The representation in rodent brains indeed has two distinct modules of non-overlapping cell populations: (1) GCs (Hafting et al., 2005) which represent allocentric space via hexagonal firing patterns (Fig. 6A); and (2) OVCs (Høydal et al., 2019) which represent relative location to objects through firing fields at specific relative distances and orientations (Fig. 6B).

**Model with additional structural constraint.** Predicting allocentric spatial locations from egocentric self-motion cues is known as path integration (Burak & Fiete, 2009), and is believed to be a fundamental function of entorhinal cortex (where GCs and OVCs are found). GCs naturally emerge from training RNNs to path integrate under several additional biological constraints (Sorscher et al., 2019; 2020; Banino et al., 2018; Cueva & Wei, 2018). Hence to model this task (with locations *and* objects) we could train an RNN, $\boldsymbol{z}$, that predicts (1) what the spatial location, $\boldsymbol{x}$, will be and (2) whether we will encounter an object, after an action, $\boldsymbol{a}$, from the current location, and (3) what the expected action, $\boldsymbol{a}$, will be. However, here we adopt a far more general framework that does not limit future applications of our approach simply to sequential integration problems.

In particular, it was recently shown (Gao et al., 2021) that path integration constraints can be applied directly on the representation by adding a new constraint in the loss imposing

$$\boldsymbol{z}(\boldsymbol{x}) = f(\boldsymbol{W_a}\boldsymbol{z}(\boldsymbol{x} - \boldsymbol{a})). \tag{3}$$

Here $f(\cdot)$ is an activation function and $\boldsymbol{W_a}$ is a weight matrix that depends on the action $\boldsymbol{a}$. This surrogate constraint imposes potential path integration by ensuring that a motion $\boldsymbol{a}$ in space $\boldsymbol{x}$ imposes a lawful change in neural representation $\boldsymbol{z}$, thereby transforming the sequential path integration problem into the problem of directly estimating neural representations of space (see Appendix A.9 for more details). Thus we minimise:

$$\mathcal{L} = \underbrace{\mathcal{L}_{\text{nonneg}} + \mathcal{L}_{\text{activity}} + \mathcal{L}_{\text{weight}}}_{\text{Biological constraints}} + \underbrace{\mathcal{L}_{\text{location}} + \mathcal{L}_{\text{actions}} + \mathcal{L}_{\text{objects}}}_{\text{Functional constraints}} + \underbrace{\mathcal{L}_{\text{path integration}}}_{\text{Structural constraints}} \tag{4}$$

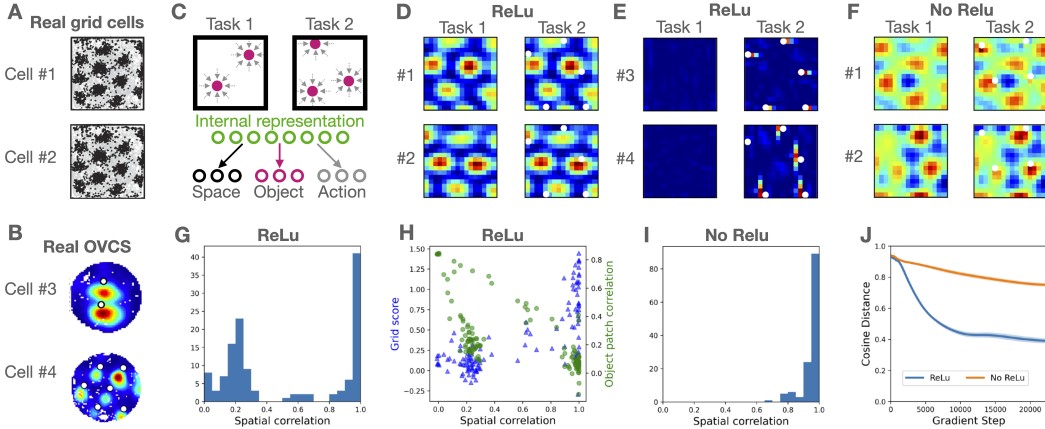

Figure 6: **Modules of distinct cell types form with nonnegativity and factorised tasks. A)** When rodents navigate environments with objects, GCs encode location in physical space with firing fields lying on a hexagonal lattice, while **B)** OVCs encode relative location to objects with firing fields at specific distances and orientations from (white) objects. These plots are ratemaps; the average firing of a given cell at every location. **C)** Top: To model these cells we use a task environment where objects move location in different contexts - space and objects are factorised. Bottom: We train a representation to predict 1) spatial location, 2) object location, and 3) correct action at every location. **D-E)** Model ratemaps when trained *with* a ReLU activation function. Task 1 contains no objects, while task 2 has several objects (white dots). We see two types of cell representation: **D)** GCs that do not change across tasks and **E)** OVCs that only appear in the presence of objects. **F)** Model ratemaps when trained *without* ReLu activation. All representations are multi-peaked but amorphous. Further cells representations shown in Fig. 14. **G)** To quantify modules in the ReLu model, we compute the distribution of individual cell spatial correlations across different tasks. **H)** The mode with high spatial correlation are cells that have high grid-score (Barry et al., 2012) and are grid cells (GCs), the mode with low spatial correlation are cells that respond similarly around objects (OVCs). **I)** Only one mode of cells is seen without a ReLu activation. **J)** To quantify module-ness over many random seeds, we compute the cosine distance between the population's contribution to $x$ and $o$. This is done by taking the absolute value of weights projecting from $z$ to $x$ and $o$, then summing over the space/object dimension (to obtain a vector the same dimension as $z$), then computing the cosine distance. We see low cosine distance for the ReLu indicating different cells code for space vs objects - i.e. modules.

These are the same biological constraints as above, but now the functional constraints involve predicting location, object, and action, and an additional structural constraint imposes equation 3. Interestingly, the structural constraint leads to a pattern forming optimisation dynamics (see Appendix A.9 for mathematical details).

**Modules of distinct cell types when tasks are factorised and representations are nonnegative.**
Just as our theory predicts, when training on tasks where objects and space are factorised (i.e. objects can be anywhere in space), under our biological constraints of nonnegativity and energy efficiency, distinct neural modules emerge, each selective for a single task factor (Fig. 6D-E). We see GC-like neurons that consistently represent space independent of object locations, and OVC-like neurons that recenter their representations around the moving objects or are inactive if no objects are present (further cells are shown in Fig. 14). Whereas without the nonnegativity constraint, all cells look qualitatively similar - a single module of multi-peaked amorphous cells (Fig. 6F). To quantify whether a population really has two distinct cell types (two modules) we analyse the consistency of a cell's representation by taking its average spatial correlation between many different object configurations (Fig. 6G-I). In the ReLu case, there are two modes (Fig. 6G-I) showing a double dissociation: the mode containing cells that don't change have high grid-score (Barry et al., 2012) and do not have consistent activity around objects. These are GCs. Whereas the mode containing cells that change between tasks have low grid-score and respond consistently around objects (Fig. 6H). These cells respond to objects and are comparable to OVCs.

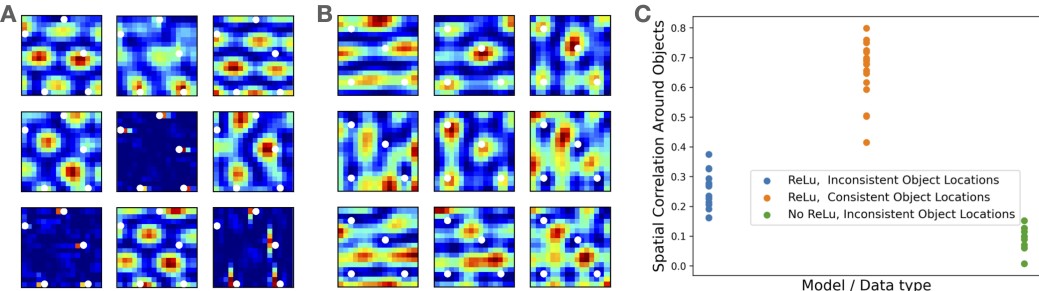

Figure 7: **Entangled tasks lead to entangled representations and grid cell warping.** We show a representative selection of cells from a model with **A)** a factorised task (as in Fig. 6) and **B)** an entangled task. The phases of the firing fields in the entangled task are locked to object location - they have warped their firing fields. This is not the case for the factorised task (asides from object specific cells). **C)** To quantify phase locking for each of the task/model variant, we compute the average spatial correlation of patches around objects. Only the entangled task shows high correlation, i.e. the cell representations have warped around objects. Each point is a model trained from a random seed.

**Grid cell warping and mixed-selectivity when tasks are entangled.** Experimental results show GCs sometimes warp their firing fields towards rewarded locations (Boccara et al., 2019) and sometimes don't (Butler et al., 2019). Intriguingly, in the warping situation, the rodents exhibited stereotyped behaviour; sequentially running between rewards using the same spatial trajectories rather than freely exploring the space (i.e. behaviour is entangled with space). We now explain these neuroscience observations as a consequence of space becoming entangled with objects/rewards.

Modelling factorised versus entangled tasks (objects changing locations versus always staying in the same locations), produces very different GC behaviours. In the factorised case, grid fields are unrelated to objects (Fig. 7A), whereas in the entangled task grid fields warp to objects (Fig. 7B). We quantify this by measuring the average spatial correlation of patches around each object. Only when the task is entangled are fields consistently warped towards objects (Fig. 7C). Thus we have an explanation for GC warping; they warp when space can no longer be disentangled from other factors (e.g. objects or behaviour) in behavioural tasks.

## 5 DISCUSSION

We have proven that simple biological constraints like nonnegativity and energy efficiency lead to disentanglement, and empirically verified this in machine learning and neuroscience tasks, leading to a new understanding of functional cell types. We now consider some more neuroscience implications.

**Representing categories.** Our theory additionally says that representations of individual categories should be encoded in separate neural populations (disentangled; see Appendix A.7 [4].). This provides a potential explanation for "grandmother cells" that selectively represent specific concepts or categories (Quiroga et al., 2005), and have long puzzled proponents of distributed representations It further potentially explains situations where animals have been trained on multiple tasks, and different neurons are found to engage in each task (Rainer et al., 1998; Roy et al., 2010; Asaad et al., 2000; Lee et al., 2022; Flesch et al., 2022). We note that while our theory says you need at least one neuron per concept that is read-out (true even without disentanglement), it does not say you need a neuron for every possible concept - only for ones that are explicitly read-out.

**When to disentangle?** Our theory speaks to situations in which brains or networks must generalise to new combinations of learnt factors. In this situation, if the input-output (or input-latent-output) transformations are linear, biological constraints will cause complete disentanglement of the networks. When the mappings are nonlinear, we show empirically that mixed-selectivity exists, but gradually de-mixes as layers approach the output (in supervised), or latent (in unsupervised), layers.

---

[4].

[4]We note this phenomena could also be accounted for with a sparsity constraint.

Optimising for low firing contrasts with previous ideas which instead optimise for linear read-out. This latter situation is akin to kernel regression, where mixed-selectivity through random expansion increases the dimensionality of neural representations, allowing simple linear read-outs in the high dimensional space to perform arbitrary nonlinear operations on the original low dimensional space.

**Mixed-selectivity.** Mixed-selectivity exists in the brain. For example, Kenyon cells in the Drosophila mushroom body increase the dimensionality of their inputs by an order of magnitude by close to random projections (Aso et al., 2014). This may allow linear read-out to behaviour via simple dopamine gating. Similarly rodent hippocampal cells encode conjunctions of spatial and sensory variables to allow rapid formation of new memories (Komorowski et al., 2009)]. More recently it has been suggested that PFC neurons have this same property, for the same reason (Rigotti et al., 2013). However, it is less clear that this is a general property of representations in associative cortex (including PFC), which can separate into different neuronal representations of different interpretable factors (Hirokawa et al., 2019; Bernardi et al., 2020) or tasks (Lee et al., 2022; Flesch et al., 2022).

One possibility is that in overtrained situations with only a relatively small number of categories or trial-types (where mixed-selectivity has been observed), the task can effectively be solved by categorising the current trial into one of a few previous experiences. By contrast in tasks where combinatorial generalisation is required, the factored solution may be preferred.

**A program to understand how brain representations structure themselves.** This work is one piece of the puzzle. It tells us when neural circuits systems should represent different factors in different neurons. It does not tell us, however, how each factor itself should be represented. For example it does not tell us why GCs and OVCs look the ways they do. We believe that the same principles of nonnegativity, minimising neural activity, and representing structure, will be essential components obtaining this more general understanding. Indeed in a companion paper, we use the same constraints, along with formalising structure/path-integration using group and representations theory, to mathematically understand why grid cells look like grid cells (Dorrell et al., 2023). Similarly, our current understanding is limited to the optimal solution for factorised representations, but we anticipate similar ideas will be applicable to neural dynamics (Driscoll et al., 2022).

## 6 CONCLUSION

We introduced constraints inspired by biological neurons - nonnegativity and energy efficiency (w.r.t. either activity or weights) - and proved these constraints lead to linear factorised codes being disentangled. We empirically verified this in simulation, and showed the same constraints lead to disentanglement with both nonlinear data and nonlinear networks. We even achieve competitive disentanglement scores on a baseline disentanglement task, even though this was not our specific aim. We showed these biological constraints explain why neuroscientists observe bespoke cell types, e.g. GCs (Hafting et al., 2005), OVCs (Høydal et al., 2019), border vector cells (Solstad et al., 2008; Lever et al., 2009), since space, boundaries, and objects appear in a factorised form (i.e. occur in any independent combination), and so are optimally represented by different neural populations for each factor. These same principles explain why neurons in inferior temporal cortex are axis aligned to underlying factors of variation that generate the data they represent (Chang & Tsao, 2017; Bao et al., 2020; Higgins et al., 2021), why visual cortex neurons are decorrelated (Ecker et al., 2010), or why neurons in parietal cortex only selective for specific tasks (Lee et al., 2022). Lastly, we also explained the confusing finding of grid fields warping towards rewards (Boccara et al., 2019) as the space and rewards becoming entangled.

This work bridges the gap between single neuron and population responses, and offers an understanding of properties of neural representations in terms of task structure above and beyond just dimensionality. Additionally it demonstrates the utility of neurobiological considerations in designing machine learning algorithms. Overall, we hope this work demonstrates the promise of a unified research program that more deeply connects the neuroscience and machine learning communities to help in their combined quest to both understand and learn neural representations. Such a unified approach spanning brains and machines could help both sides, offering neuroscientists a deeper understanding of how cortical representations structure themselves, and offering machine learners novel ways to control and understand the representations their machines learn.

ACKNOWLEDGEMENTS

We thank Emile Mathieu and Andrew Saxe for helpful comments and advice on our manuscript. We thank the following funding sources: Sir Henry Wellcome Post-doctoral Fellowship (222817/Z/21/Z) to J.C.R.W.; the Gatsby Charitable Foundation to W.D.; the James S. McDonnell, Simons Foundations, NTT Research, and an NSF CAREER Award to S.G.; Wellcome Principal Research Fellowship (219525/Z/19/Z), Wellcome Collaborator award (214314/Z/18/Z), and Jean-François and Marie-Laure de Clermont-Tonnerre Foundation award (JSMF220020372) to T.E.J.B.; the Wellcome Centre for Integrative Neuroimaging and Wellcome Centre for Human Neuroimaging are each supported by core funding from the Wellcome Trust (203139/Z/16/Z, 203147/Z/16/Z).

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

# A APPENDIX

## A.1 DEFINITIONS

**Disentanglement definition**. We define disentanglement as when single *model* neurons care about single ground truth factors. We do not mind if more than one model neuron cares about the same factor. We note that other definitions of disentanglement are when *single* ground truth factors of variation are encoded in at most a *single* model neuron - we do not ask for this level of parsimony.

**Disentanglement metric**. Our metric (mutual information ratio; MIR) measures the mutual information between neurons and factors, $\mathrm{I}_{n,f}$, then calculates each neuron's preference (we exclude inactive neurons) for one factor over the rest via

$$r_n = \frac{\max_f(\mathrm{I}_{n,f})}{\sum_f \mathrm{I}_{n,f}} \tag{5}$$

We then divide by the number of (active) neurons, $n_n$, and normalise for the number of factors, $n_f$, i.e.

$$MIR = \frac{\frac{\sum_n r_n}{n_n} - \frac{1}{n_f}}{1 - \frac{1}{n_f}} \tag{6}$$

High MIR means single neurons show high preference for a single ground truth factor.

We do not propose this metric as a gold-standard metric for disentanglement, indeed in some situations such as spiral shaped functions, it could exhibit pathological properties. Furthermore since our metric is normalised by the sum mutual information of all factors, rather than the just the gap between the factors with 1st and 2nd highest MI (like the MIG metric), it appears lower even when with near perfect disentanglement. However in our situations it works just fine. Again, we use it over other metrics since other metrics penalise situations where more than one neuron codes for a ground-truth factor.

## A.2 RELATED WORK

Here we review the literature on disentanglement and the use of biological constraints, in both machines and in the brain, and where possible relate back to the work presented in this paper.

While there are many technique for learning linear features of data, by far the most relevant to this work is independent component analysis (ICA), though we briefly mention principal component analysis (PCA) here too. ICA seeks to learn source signals in data that are maximally independent (Comon, 1994), and there exist algorithms (Hyvärinen & Oja, 2000) that provably extract these components when the underlying sources are linearly mixed in the observed data (provided the sources are non-gaussian). PCA, on the other hand, aims to find the minimally correlated components of the signal that explain the most variance, and simple algorithms exist for PCA too.

### A.2.1 LINEAR DISENTANGLEMENT

PCA can be done in neural hardware too. Oja's rule (Oja, 1982)) is a simple learning rule that learns principle components sequentially. More recently, neural algorithms for learning the principal components non-sequentially have been developed, either using Hebbian/anti-Hebbian learning rules in biological networks (Pehlevan et al., 2015) or specific weight regularization in linear autoencoders (Kunin et al., 2019).

Neural algorithms for ICA also exist. These draw on ideas from nonnegative ICA (Plumbley, 2002; 2003) - where the sources are assumed to be nonnegative - and use a similarity matching objective to derive biologically plausible networks for ICA (Lipshutz et al., 2022; Pehlevan et al., 2017), with this extendable to correlated sources (Bozkurt et al., 2022). These works relate to ours as they consider nonnegativity, however their focus is on biological neural networks implementing *linear* ICA, whereas we mathematically prove biological constraints including nonnegativity lead to source separation in linear ICA, and empirically show that the same is true from *nonlinear* ICA.

### A.2.2 NONLINEAR DISENTANGLEMENT

There exist no known algorithm for *exact* source recovery ('identifying' the sources) when the factors are nonlinearly mixed. Indeed it is provably the case that nonlinear ICA is not identifiable (Hyvärinen et al., 1999).

Nevertheless, there have many methods for disentanglement in *nonlinear* models, with most modern methods use variational autoencoders (VAE; Kingma & Welling (2013)), with various choices that promote explicit factorisation of the learned latent space. For example $\beta$-VAEs (Higgins et al., 2017a) up-weight (by $\beta$) the term in the VAE loss that encourages the posterior to be a factorised distribution. Other variations of disentanglement VAEs (Burgess et al., 2018; Kim & Mnih, 2018; Ridgeway & Mozer, 2018; Kumar et al., 2018; Chen et al., 2018) similarly try to explicitly enforce a factorised aggregate posterior.

Importantly, analogously to the nonlinear ICA case, it has been shown that disentanglement is not possible without inductive bias in the data or model (Locatello et al., 2019), and it has been shown that generic VAEs are only able to disentangle when there is a particular bias in the data (Rolinek et al., 2019; Zietlow et al., 2021). However, it has also been shown that numerous inductive biases are sufficient for disentanglement (i.e. conditioning the prior on a third variable Khemakhem et al. (2020); or local isometry and non-Gaussianity Horan et al. (2021)).

Though, in our work, we show disentanglement in VAEs with biological constraints, we also show disentanglement in supervised DeepNets and linear and non-linear autoencoders. Empirically, it seems out constraints are not limited to the VAE case, though further and more intensive empirical investigation are needed to confirm this.

Our work also relates to modern self-supervised learning (Bardes et al., 2022) which seeks decorrelated representations with non-zero variance. Similar to many of the VAEs that disentangle, they introduce loss terms specifically to force decorrelation. Here instead, we show that simple biological constraints of nonnegativity and energy efficiency lead to emergent disentanglement without an explicit decorrelation term in the objective.

There is already evidence of disentanglement in machines as an emergent phenomenon as opposed to being baked in, for example interpretable neurons in vision model (Bau et al., 2017), multi-modal neurons in CLIP (Goh et al., 2021), and other work by on superposition/ privileged axes (Elhage et al., 2022) that also discusses the role of nonnegativity constraints. Similarly, other work is on understanding specialization in DeepNets (Casper et al., 2022).

### A.2.3 Disentanglement in brains

Barlow first hypothesised that neurons represent independent components via redundancy reduction - minimising mutual information between their representation (Barlow, 1961; 1972) [5]. Since Barlow there have been numerous experimental results suggesting that brain neurons are disentangled: in inferior temporal cortex (Chang & Tsao, 2017; Bao et al., 2020; Higgins et al., 2021; Yildirim et al., 2020), visual cortex (Ecker et al., 2010; Gáspár et al., 2019), parietal cortex (Lee et al., 2022), and entorhinal cortex (Hafting et al., 2005; Solstad et al., 2008; Høydal et al., 2019). Similarly, however, there have been many reported neurons in various brain regions that are apparently mixed-selective (Rigotti et al., 2013; Boccara et al., 2019; Butler et al., 2019; Komorowski et al., 2009).

There is no formal understanding of when and why neurons in the brain factorise across task parameters. Nevertheless, single latent dimensions from disentanglement models do predict single neuron activity, implying biological neurons are disentangled (Higgins et al., 2021). One conjecture in neuroscience is that while representing task factors is important for generalisation, mixed-selectivity is important for efficient read-out (Behrens et al., 2018; Rigotti et al., 2013; Bernardi et al., 2020). Here we show disentangled brain representations are preferred if the task is factorised into independent factors. This work bridges the gap between a single neuron understanding (i.e. like Sherrington) and a population based understanding (i.e. like Hopfield) (Barack & Krakauer, 2021).

### A.2.4 Biological constraints

While these constraints of nonnegativity and energy efficiency are not new to the machine learning or neuroscience literature, we package them together to learn new insights into how biological constraints shape the space of neural representation.

**Nonnegativity.** Recent work on multitask learning in recurrent neural networks (RNNs) (Yang et al., 2019; Driscoll et al., 2022) demonstrated that neural populations, with a nonnegative activation function, partition themselves into task specific modules (Driscoll et al., 2022). Nonnegativity is also important in obtaining hexagonal, not square, grid cells (Dordek et al., 2016; Whittington et al., 2021b; Sorscher et al., 2019; 2020). Also, nonnegative matrix factorisation empirically yields spatially localised factors for images (Lee & Seung, 2000). Other work discusses potential algorithmic functions of nonnegativity in again Hebbian/anti-Hebbian networks, relating it to clustering, sparse feature discovery and again ICA (Pehlevan et al., 2015). Our work demonstrates theoretically and empirically *why* nonnegativity leads to single neurons becoming selective for single factors.

**Energy efficiency.** Using regularisation is common in machine learning, with l2 or l1 losses on weights primarily used. Less commonly do machine learners ask for energy efficiency on activations, though the VAE loss does include energy efficiency latent activations, and additionally some activation functions can be see as energy minimising, e.g. ReLus are sparsity inducing. In neuroscience, there is much work considering energy efficiency (see Chintaluri & Vogels (2022) for a fun recent example). There are also results more relevant to us, for example work that test whether orientation of neural manifolds from mouse V1 minimise energy efficiency (spike count) by randomly rotating and optimally shifting them to fit into nonnegative orthants (Bordelon & Pehlevan, 2022).

---

[5]We note that Barlow also considered parsimony as a representational pressure too. While we do not consider it in this paper, we believe it is an important constraint.

## A.3 DETAILS OF DERIVATIONS

### A.3.1 PROOF OF CONSTRAINTS LEADING TO DISENTANGLEMENT FOR A GIVEN POPULATION VARIANCE.

**Theorem.** Let $e \in \mathbb{R}^k$ be a random vector whose $k$ independent components denote $k$ task factors. We assume each independent task factor $e_i$ is drawn from a distribution that has mean 0, variance $\sigma^2$, and maximum and minimum values of $\min(e_i) = -a$ and $\max(e_i) = a$. Also let $z \in \mathbb{R}^n$ be a linear neural representation of the task factors given by

$$z = Me + b_z, \tag{7}$$

where $M \in \mathbb{R}^{n \times k}$ are mixing weights and $b_z \in \mathbb{R}^n$ is a bias. We further assume two constraints: (1) the neural representation is *nonnegative* with $z_i \geq 0$ for all $i = 1, \ldots, n$, and (2) the neural population variance is a nonzero constant, $\sum_j Var(z_j) = C$, so that the neural representation retains some information about the task variables. Under these two constraints we show that in the space of all possible neural representations (parameterised by $M$ and $b_z$), the representations that achieve minimal activity energy $\mathbb{E}||z||^2$ also exhibit disentanglement, by which we mean every neuron $z_j$ is selective for at most one task parameter: i.e. $|M_{jk}||M_{jl}| = 0$ for $k \neq l$.

**Proof.** We aim to find a representation, $z$, that minimises activity energy (the expected norm of $z$), is nonnegative, all for a fixed population variance, $C$. This is a constrained optimisation problem, and equates to understanding what $M$ and $b_z$ must look like in order to satisfy our constraints, and minimise $\mathbb{E}||z||^2$.

$$\underset{M, b_z}{\text{minimise}} \, \mathbb{E}||z||^2 \quad \text{s.t. } z_i \geq 0 \,, \; \sum_j Var(z_j) = C \tag{8}$$

The total activity energy, i.e. the expected norm of $z$, is

$$\begin{aligned} \mathbb{E}||z||^2 = \sum_j \mathbb{E}(z_j^2) &= \sum_j Var(z_j) + (\mathbb{E}z_j)^2 \\ &= \sum_j Var((Me)_j) + \sum_j ((b_z)_j)^2 \end{aligned} \tag{9}$$

We want to minimise this, under the constraint $z_j \geq 0$. To satisfy this constraint, $(b_z)_j$ must account for any negativity. This means that

$$(b_z)_j \geq -\min(Me)_j = \sum_k |M_{jk}|a \tag{10}$$

Where the last equality is because all $e_k$ are i.i.d. and so the minimum of a sum of random variables is the sum of their minima. The modulus sign is since $M_{jk}$ can be positive or negative, so we need to consider the maximum and minimum of $e_k$, and an assumption of ours was that the maximum and minimum had the same value $a$. Thus

$$(b_z)_j = \sum_k |M_{jk}|a + p_j \tag{11}$$

Where $p_j \geq 0$. Intuitively, $p_j = 0$ since anything else would increase activity energy. Formally,

$$\begin{aligned} \mathbb{E}||z||^2 &= \sum_j Var((Me)_j) + \sum_j ((b_z)_j)^2 \\ &= \sigma^2 \sum_{j,k} M_{jk}^2 + \sum_j (\sum_k |M_{jk}|a + p_j)^2 \\ &= \sigma^2 \sum_{j,k} M_{jk}^2 + \sum_j (\sum_k |M_{jk}|a)^2 + p_j^2 + 2p_j^2 \sum_k |M_{jk}|a \end{aligned} \tag{12}$$

Since both $\sum_k |M_{jk}|a$ and $p_j$ are greater than zero, $p_j$ always increases activity energy, no matter what $M_{jk}$ is. Thus we set it to zero, i.e. $p_j = 0$. Hence we can simplify the expected activity as

follows

$$
\begin{aligned}
\mathbb{E}||\boldsymbol{z}||^2 &= \sigma^2 \sum_{j,k} M_{jk}^2 + \sum_j (\sum_k |M_{jk}|a)^2 \\
&= \sigma^2 \sum_{j,k} M_{jk}^2 + \sum_j (\sum_k |M_{jk}|a)(\sum_l |M_{jl}|a) \\
&= \sigma^2 \sum_{j,k} M_{jk}^2 + a^2 \sum_{j,k}(M_{jk}^2 + \sum_{l \neq k} |M_{jk}||M_{jl}|) \\
&= (1 + \frac{a^2}{\sigma^2}) \sum_j Var(z_j) + a^2 \sum_{j,k,l \neq k} |M_{jk}||M_{jl}| \\
&= (1 + \frac{a^2}{\sigma^2})C + a^2 \sum_{j,k,l \neq k} |M_{jk}||M_{jl}|
\end{aligned}
\tag{13}
$$

Now all the constraints have been incorporated, our only job is to minimise $\mathbb{E}||\boldsymbol{z}||^2$. This is done when $\sum_{j,k,l \neq k} |M_{jk}||M_{jl}| = 0$, and that only happens when $|M_{jk}||M_{jl}| = 0$ for all $j$ and $l \neq k$. In words, this means that neuron $j$ in only receives information from one element of $\boldsymbol{e}$. This is disentanglement.

### A.3.2 PROOF THAT POPULATION VARIANCE IS BOUNDED BY READ-OUT WEIGHTS NORM

**Theorem.** Let $\boldsymbol{x} = \boldsymbol{D}\boldsymbol{e}$ be observed entangled data, where $\boldsymbol{x} \in \mathbb{R}^m$, $\boldsymbol{D} \in \mathbb{R}^{m \times k}$, and $\boldsymbol{e} \in \mathbb{R}^k$ is a random vector whose $k$ independent components denote $k$ task factors. We assume each independent task factor $e_i$ is drawn from a distribution that has mean 0, variance $\sigma^2$, and maximum and minimum values of $\min(e_i) = -a$ and $\max(e_i) = a$. Let a neural representation $\boldsymbol{z} \in \mathbb{R}^n$ exactly predict observed data via $\boldsymbol{x} = \boldsymbol{W}\boldsymbol{z} + \boldsymbol{b}_x$ with zero error, i.e. $\boldsymbol{x} = \boldsymbol{D}\boldsymbol{e} = \boldsymbol{W}\boldsymbol{z} + \boldsymbol{b}_x$. Where $\boldsymbol{W} \in \mathbb{R}^{m \times n}$ $m \geq n \geq k$, are read-out weights and $\boldsymbol{b}_x \in \mathbb{R}^m$ is an offset.

Then for all such data generation models (with parameters $\boldsymbol{D}$) and all such neural representations (with parameters $\boldsymbol{W}$ and $\boldsymbol{b}_x$), as long as: (1) the smallest singular value of $\boldsymbol{D}$ is non-zero, $\sigma_{min}(\boldsymbol{D}) > 0$; (2) the norm of the read-out weights $||\boldsymbol{W}||_F^2$ is finite; then the population variance of $\boldsymbol{z}$ is bounded from below by the norm of the read-out weights.

$$
\sum_j Var(z_j) \geq k^2 \frac{\sigma_{min}^2(\boldsymbol{D})}{||\boldsymbol{W}||_F^2}
\tag{14}
$$

**Proof.** Since $\boldsymbol{x} = \boldsymbol{W}\boldsymbol{z} + \boldsymbol{b}_x$, we have

$$
\begin{aligned}
\boldsymbol{z} &= \boldsymbol{W}^+ \boldsymbol{x} - \boldsymbol{W}^+ \boldsymbol{b}_x \\
&= \boldsymbol{W}^+ \boldsymbol{D}\boldsymbol{e} - \boldsymbol{W}^+ \boldsymbol{b}_x
\end{aligned}
\tag{15}
$$

Where $\boldsymbol{W}^+$ is the Moore-Penrose pseudoinverse. Additionally since the read-out error is zero, $\boldsymbol{W}$ must contain all the same information as $\boldsymbol{D}$, and so must be $\boldsymbol{W} = \boldsymbol{D}\boldsymbol{F}^+$, where $\boldsymbol{F}^+ \in \mathbb{R}^{k \times n}$ is a matrix with rank $k$ ($n \geq k$). The pseudoinverse of $\boldsymbol{W}$ is thus $\boldsymbol{W}^+ = \boldsymbol{F}\boldsymbol{D}^+$. The population variance becomes

$$
\begin{aligned}
\sum_j Var(z_j) &= Tr(Var(\boldsymbol{W}^+ \boldsymbol{D}\boldsymbol{e} - \boldsymbol{W}^+ \boldsymbol{b}_x)) \\
&= Tr(Var(\boldsymbol{F}\boldsymbol{D}^+ \boldsymbol{D}\boldsymbol{e})) \\
&= Tr(Var(\boldsymbol{F}\boldsymbol{e})) \\
&= Tr(\boldsymbol{F}Var(\boldsymbol{e})\boldsymbol{F}^T) \\
&= \sigma^2 Tr(\boldsymbol{F}\boldsymbol{F}^T) \\
&= \sigma^2 ||\boldsymbol{F}||_F^2
\end{aligned}
\tag{16}
$$

Using the fact that the norm of a matrix's pseudoinverse is bounded by the norm of the matrix

$$
||\boldsymbol{F}^+||_F^2 = Tr((\boldsymbol{F}^T\boldsymbol{F})^{-1}) \geq \frac{k^2}{||\boldsymbol{F}||_F^2}
\tag{17}
$$

Along with using the following trace identity (Fang et al., 1994)

$$||\boldsymbol{F}^+||_F^2 \sigma_{min}^2(\boldsymbol{D}) \leq ||\boldsymbol{W}||_F^2 = ||\boldsymbol{D}\boldsymbol{F}^+||_F^2 \leq ||\boldsymbol{F}^+||_F^2 \sigma_{max}^2(\boldsymbol{D}) \tag{18}$$

Where $\sigma_{min}(\boldsymbol{D})$ and $\sigma_{max}(\boldsymbol{D})$ are the smallest and largest singular values of $\boldsymbol{D}$. Thus

$$\frac{1}{||\boldsymbol{F}^+||_F^2} \geq \frac{\sigma_{min}^2(\boldsymbol{D})}{||\boldsymbol{W}||_F^2} \tag{19}$$

Combining it all together

$$\begin{aligned}
\sum_j Var(z_j) &= \sigma^2 ||\boldsymbol{F}||_F^2 \\
&\geq \frac{k^2 \sigma^2}{||\boldsymbol{F}^+||_F^2} \\
&\geq \frac{k^2 \sigma^2 \sigma_{min}^2(\boldsymbol{D})}{||\boldsymbol{W}||_F^2}
\end{aligned} \tag{20}$$

We note that the first inequality becomes an equality if and only if $\boldsymbol{F}^+$ has (scaled) orthonormal *rows*, and the second inequality becomes and equality if and only if $\boldsymbol{D}$ has (scaled) orthonormal *columns*.

### A.3.3    PROOF OF DISENTANGLEMENT WHEN DATA GENERATIVE MATRIX HAS (SCALED) ORTHONORMAL COLUMNS

**Theorem**. Let $\boldsymbol{x} = \boldsymbol{D}\boldsymbol{e}$ be observed entangled data, where $\boldsymbol{D} \in \mathbb{R}^{m \times k}$, and $\boldsymbol{e} \in \mathbb{R}^k$ is a random vector whose $k$ independent components denote $k$ task factors. We assume each independent task factor $e_i$ is drawn from a distribution that has mean 0, variance $\sigma^2$, and maximum and minimum values of $\min(e_i) = -a$ and $\max(e_i) = a$. Let a neural representation $\boldsymbol{z} \in \mathbb{R}^n$ exactly predict observed data via $\boldsymbol{x} = \boldsymbol{W}\boldsymbol{z} + \boldsymbol{b}_x$ with zero error, i.e. $\boldsymbol{x} = \boldsymbol{D}\boldsymbol{e} = \boldsymbol{W}\boldsymbol{z} + \boldsymbol{b}_x$. Where $\boldsymbol{W} \in \mathbb{R}^{m \times n}$ $m \geq n \geq k$, are read-out weights and $\boldsymbol{b}_x \in \mathbb{R}^m$ is an offset.

Then for all such data generation models (with parameters $\boldsymbol{D}$) and all such neural representations (with parameters $\boldsymbol{W}$ and $\boldsymbol{b}_x$), as long as: (1) the columns of $\boldsymbol{D}$ are (scaled) orthonormal; (2) the norm of the read-out weights $||\boldsymbol{W}||_F^2$ is finite; (3) the neural representation is nonnegative (i.e. $\boldsymbol{z} > 0$), then out of all such neural representations, the minimum energy representations are also disentangled ones. By this we mean that each neuron $z_i$ will be selective for at most one hidden task factor $e_j$.

**Proof.** We aim to find a representation, $\boldsymbol{z}$, that minimises activity energy (the expected norm of $\boldsymbol{z}$), is nonnegative, and predicts data $\boldsymbol{x} = \boldsymbol{D}\boldsymbol{e}$ via $\boldsymbol{x} = \boldsymbol{W}\boldsymbol{z} + \boldsymbol{b}_x$ with zero error ($\boldsymbol{D}\boldsymbol{e} = \boldsymbol{W}\boldsymbol{z} + \boldsymbol{b}_x$). This is a constrained optimisation problem, and equates to understanding what $\boldsymbol{W}$ and $\boldsymbol{b}_x$ (and therefore $\boldsymbol{z}$) must look like in order to satisfy our constraints, and minimise $\mathbb{E}||\boldsymbol{z}||^2$.

$$\underset{\boldsymbol{W},\boldsymbol{b}_x}{\text{minimise}} \, \mathbb{E}||\boldsymbol{z}||^2 \quad \text{s.t. } z_i \geq 0 \, , \, \boldsymbol{D}\boldsymbol{e} = \boldsymbol{W}\boldsymbol{z} + \boldsymbol{b}_x \, \forall \boldsymbol{e} \, , \, ||\boldsymbol{W}||_F^2 = K \tag{21}$$

We note that this is the classic matrix factorisation setting. The difference here is that we ask the representation $\boldsymbol{z}$ to be nonnegative.

Firstly, we note that since the columns of $\boldsymbol{D}$ are (scaled) orthonormal, then the singular values are all equal: $\sigma_{min}(\boldsymbol{D}) = \sigma_{max}(\boldsymbol{D}) = \sigma(\boldsymbol{D})$. This result is useful as now the inequality in equation 19 becomes an equality

$$\frac{1}{||\boldsymbol{F}^+||_F^2} = \frac{\sigma^2(\boldsymbol{D})}{||\boldsymbol{W}||_F^2} \tag{22}$$

Now we prove disentanglement. Since the read-out error is zero, $\boldsymbol{W}$ must contain all the same information as $\boldsymbol{D}$ and so must be $\boldsymbol{W} = \boldsymbol{D}\boldsymbol{F}^+$. Using this, and repeating a similar process to the

first proof (Appendix A.3.1), we get

$$
\begin{aligned}
\mathbb{E}||\boldsymbol{z}||^2 &= (1 + \frac{a^2}{\sigma^2}) \sum_j Var(z_j) + a^2 \sum_{j,k,l \neq k} |(\boldsymbol{W}^+\boldsymbol{D})_{jk}||(\boldsymbol{W}^+\boldsymbol{D})_{jl}| \\
&= (\sigma^2 + a^2)||\boldsymbol{F}||_F^2 + a^2 \sum_{j,k,l \neq k} |F_{jk}||F_{jl}| \\
&\geq (\sigma^2 + a^2)\frac{k^2}{||\boldsymbol{F}^+||_F^2} + a^2 \sum_{j,k,l \neq k} |F_{jk}||F_{jl}| \\
&= (\sigma^2 + a^2)\frac{k^2\sigma^2(\boldsymbol{D})}{||\boldsymbol{W}||_F^2} + a^2 \sum_{j,k,l \neq k} |F_{jk}||F_{jl}|
\end{aligned}
\tag{23}
$$

Where the inequality is due to equation 17. This inequality becomes an equality if and only if $\boldsymbol{F}^+$ has (scaled) orthonormal *rows* (i.e. $\boldsymbol{F}$ has (inverse scaled) orthonormal *columns*). This means that for a fixed $||\boldsymbol{W}||_F^2$, the first term in equation 23 is minimised when $\boldsymbol{F}^+$ has (scaled) orthonormal rows, or equally when $\boldsymbol{F}$ has (inverse scaled) orthonormal columns. This is interesting to us, as we can additionally make the second term in equation 23 go to zero when $\boldsymbol{F}$ is a *particular* type of matrix with (inverse scaled) orthonormal columns. Thus we will have fully optimised equation 23, and this will be our solution. First, rewriting $\boldsymbol{F}$ as a matrix with (inverse scaled) orthonormal columns

$$
\boldsymbol{F} = \frac{1}{\alpha}\boldsymbol{O} \quad \text{and} \quad \boldsymbol{F}^+ = \alpha\boldsymbol{O}^T
\tag{24}
$$

Where $\boldsymbol{O}$ is a matrix with orthonormal columns. The particular $\boldsymbol{F}$ that minimises the second term in equation 23, is when $\boldsymbol{O}$ only has, at most, a single non-zero element per row as this sets $|F_{jk}||F_{jl}| = 0$. This is disentanglement and is easily seen by

$$
\begin{aligned}
\boldsymbol{z} &= \boldsymbol{W}^+\boldsymbol{x} - \boldsymbol{W}^+\boldsymbol{b}_x \\
&= \boldsymbol{W}^+\boldsymbol{D}\boldsymbol{e} - \boldsymbol{W}^+\boldsymbol{b}_x \\
&= \alpha\boldsymbol{O}\boldsymbol{D}^+\boldsymbol{D}\boldsymbol{e} - \alpha\boldsymbol{O}\boldsymbol{D}^+\boldsymbol{b}_x \\
&= \alpha\boldsymbol{O}\boldsymbol{e} - \alpha\boldsymbol{O}\boldsymbol{D}^+\boldsymbol{b}_x
\end{aligned}
\tag{25}
$$

Since $\boldsymbol{O}$ only has a single non-zero element per row, then each $\boldsymbol{z}_i$ must only contain a single element (random variable) from $\boldsymbol{e}$. We note that nonnegativity is easily achieved by $\boldsymbol{b}_x$ learning to take a value that satisfies nonnegativity, i.e. $\alpha\boldsymbol{O}^T\boldsymbol{D}^+\boldsymbol{b}_x = \min \alpha\boldsymbol{O}^T\boldsymbol{e}$. Since $\boldsymbol{O}$ and $\boldsymbol{D}$ are full ($k$) rank, this is always possible.

We also offer an alternate proof in Appendix A.3.5 when analysing 'Simplification 1'.

### A.3.4 PROOF THAT A TALL RANDOM MATRIX WITH FINITE WIDTH HAS (SCALED) APPROXIMATELY ORTHONORMAL COLUMNS

**Theorem.** Let $\boldsymbol{D} \in \mathbb{R}^{m \times k}$ be a random matrix with elements $D_{ij}$ that are i.d.d. with expectation 0 and variance $\kappa^2/m^2$). Then as $m \to \infty$, with finite $k$, $\boldsymbol{D}^T\boldsymbol{D} \to \boldsymbol{I}$.

**Proof.**

$$
\begin{aligned}
\lim_{m \to \infty}(\boldsymbol{D}^T\boldsymbol{D})_{ik} &= \lim_{m \to \infty} \sum_j D_{ij}^T D_{jk} \\
&= m^2\mathbb{E}D_{ji}D_{jk} \\
&= \begin{cases} \kappa^2, & i = k \\ 0, & i \neq k \end{cases}
\end{aligned}
\tag{26}
$$

Thus $\boldsymbol{D}$ has orthonormal columns, scaled by $\kappa$, and so its singular values are all identical; $\sigma_{min}(\boldsymbol{D}) = \sigma_{max}(\boldsymbol{D}) = \kappa^2$.

### A.3.5 WHEN THE DATA GENERATIVE MATRIX **DOES NOT** HAVE (SCALED) ORTHONORMAL COLUMNS

While we do not prove the general case, we offer some intuition here that suggest disentangled representations are favoured in many situations. Consider the general setting in which $\boldsymbol{D} \in \mathbb{R}^{m \times k}$

has the following singular value decomposition (SVD):

$$\boldsymbol{D} = \boldsymbol{U}\Sigma\boldsymbol{V} \tag{27}$$

Where $\Sigma \in \mathbb{R}^{m \times k}$ is a rectangular diagonal matrix with positive entries, and $\boldsymbol{U} \in \mathbb{R}^{m \times m}$ and $\boldsymbol{V} \in \mathbb{R}^{k \times k}$ are orthogonal matrices. As in the above proofs, since $\boldsymbol{x}$ is predicted with zero error from $\boldsymbol{z}$, via

$$\boldsymbol{x} = \boldsymbol{D}\boldsymbol{e} = \boldsymbol{W}\boldsymbol{z} + \boldsymbol{b}_x \tag{28}$$

Thus $\boldsymbol{W}$ must be of the form $\boldsymbol{W} = \boldsymbol{D}\boldsymbol{F}^+$, where $\boldsymbol{F}^+ \in \mathbb{R}^{k \times n}$ is a rank $k$ matrix, since $\boldsymbol{z} \in \mathbb{R}^n$, $\boldsymbol{x} \in \mathbb{R}^m$, and $\boldsymbol{e} \in \mathbb{R}^k$, $m \geq n \geq k$. We define the SVD of $\boldsymbol{F}$ as

$$\boldsymbol{F} = \boldsymbol{O}\Lambda\boldsymbol{R}^T \quad , \quad \boldsymbol{F}^+ = \boldsymbol{R}\Lambda^{-1}\boldsymbol{O}^T \tag{29}$$

Where $\Lambda \in \mathbb{R}^{k \times n}$ is a rectangular diagonal matrix with positive entries, $\lambda_i$, $\Lambda^{-1} \in \mathbb{R}^{n \times k}$ is a rectangular diagonal matrix with positive entries, $\frac{1}{\lambda_i}$, and where $\boldsymbol{O} \in \mathbb{R}^{n \times n}$ and $\boldsymbol{R} \in \mathbb{R}^{k \times k}$ are orthogonal matrices. From equation 16 the population variance is

$$\sum_j Var(z_j) = \sigma^2 ||\boldsymbol{F}||_F^2$$
$$= \sigma^2 \sum_i \lambda_i^2 \tag{30}$$

Where $\sigma$ is the variance of the random variables $e_i$. The norm of the weights $||\boldsymbol{W}||_F^2$ is

$$\begin{aligned} ||\boldsymbol{W}||_F^2 &= ||\boldsymbol{D}\boldsymbol{F}^+||_F^2 \\ &= ||\boldsymbol{U}\Sigma\boldsymbol{V}^T\boldsymbol{R}\Lambda^{-1}\boldsymbol{O}^T||_F^2 \\ &= ||\Sigma\boldsymbol{V}^T\boldsymbol{R}\Lambda^{-1}||_F^2 \\ &= \sum_{ij} \frac{\sigma_i^2(D)}{\lambda_j^2} \sum_{kl} V_{ki}V_{li}R_{kj}R_{lj} \end{aligned} \tag{31}$$

While we have been previously interested in finding the minimal activity energy with fixed $||\boldsymbol{W}||_F^2$, we instead allow $||\boldsymbol{W}||_F^2$ to change and instead minimise $||\boldsymbol{W}||_F^2$: $\mathcal{L} = \mathbb{E}||\boldsymbol{z}||^2 + \beta_w||\boldsymbol{W}||_F^2$, where $\beta_w$ is the strength of regularization on the weights. Thus using equation 23, we have

$$\mathcal{L} = (\sigma^2 + a^2)\sum_i \lambda_i^2 + a^2 \sum_{j,k,l \neq k} |F_{jk}||F_{jl}| + \beta_w \sum_{ij} \frac{\sigma_i^2(D)}{\lambda_j^2} \sum_{kl} V_{ki}V_{li}R_{kj}R_{lj} \tag{32}$$

This is the overall thing we want to optimise. However, for now we do not consider the the middle term - the interaction induced by the nonnegativity constraint - and instead consider the following optimisation problem

$$\underset{\boldsymbol{V},\boldsymbol{R},\Lambda}{\text{minimise}}\, \mathcal{L}' = (\sigma^2 + a^2)\sum_i \lambda_i^2 + \beta_w \sum_{ij} \frac{\sigma_i^2(D)}{\lambda_j^2} \sum_{kl} V_{ki}V_{li}R_{kj}R_{lj} \tag{33}$$

Under the constraints that $\boldsymbol{V}$ and $\boldsymbol{R}$ remain orthogonal and $\Lambda$ remains rectangular diagonal with positive entries. This is difficult in general, but we can simplify to gain intuition and insights.

**Simplification 1**. The first simplification is when $\Sigma$ is a scaled identity matrix. This is the same simplification we used in Proof 3 (Appendix A.3.3), i.e. the singular values of $\boldsymbol{D}$ are all equal, $\sigma_i^2(D) = \sigma^2(D)$.

**Simplification 2**. The second simplification is that $\boldsymbol{V}$ is the identify matrix. This corresponds to data being generated via $\boldsymbol{D} = \boldsymbol{U}\Sigma$, i.e. the random variables are scaled and then orthogonally projected.

Analysing **simplification 1** first, to build up intuition (and offer an alternative proof of Theorem 3), $\mathcal{L}'$ becomes

$$
\begin{aligned}
\mathcal{L}' &= (\sigma^2 + a^2) \sum_i \lambda_i^2 + \beta_w \sum_{ij} \frac{\sigma_i^2(D)}{\lambda_j^2} \sum_{kl} V_{ki} V_{li} R_{kj} R_{lj} \\
&= (\sigma^2 + a^2) \sum_i \lambda_i^2 + \beta_w \sigma^2(D) \sum_j \frac{1}{\lambda_j^2} \sum_{kl} R_{kj} R_{lj} \sum_i V_{ki} V_{li} \\
&= (\sigma^2 + a^2) \sum_i \lambda_i^2 + \beta_w \sigma^2(D) \sum_j \frac{1}{\lambda_j^2} \sum_k R_{kj}^2 \\
&= (\sigma^2 + a^2) \sum_i \lambda_i^2 + \beta_w \sigma^2(D) \sum_j \frac{1}{\lambda_j^2}
\end{aligned}
\tag{34}
$$

Where we exploited the fact that $\boldsymbol{R}$ and $\boldsymbol{V}$ are orthogonal matrices. Optimising this is easy to do, we can simply take derivatives to get

$$
\lambda_i^4 = \frac{\beta_w \sigma^2(D)}{\sigma^2 + a^2}
\tag{35}
$$

This result is independent of $\boldsymbol{R}$ and $\boldsymbol{O}$, and so we are free to choose whatever orthogonal matrices we like. This is good news for us as the real game we are in is minimising $\mathcal{L}$ (equation 32) which contains an additional term involving $\boldsymbol{R}$ and $\boldsymbol{O}$: $a^2 \sum_{j,k,l \neq k} |F_{jk}||F_{jl}| = a^2 \sum_{j,k,l \neq k} |(\boldsymbol{O \Lambda R}^T)_{jk}||(\boldsymbol{O \Lambda R}^T) jl|$. Thus if we can set $|(\boldsymbol{O \Lambda R}^T)_{jk}||(\boldsymbol{O \Lambda R}^T) jl| = 0$, then we will have fully minimised $\mathcal{L}$. This is easy enough to do (keeping $\boldsymbol{R}$ and $\boldsymbol{O}$ orthogonal) and is achieved when $\boldsymbol{R}$ is a permutation matrix, and $\boldsymbol{O}$ has at most one non-zero element per row (or equally $\boldsymbol{O}$ has at most one non-zero element per column). This corresponds to disentangled representations.

Returning to **simplification 2**, where $\boldsymbol{V}$ is the identify matrix, now $\mathcal{L}'$ becomes:

$$
\begin{aligned}
\mathcal{L}' &= (\sigma^2 + a^2) \sum_i \lambda_i^2 + \beta_w \sum_{ij} \frac{\sigma_i^2(D)}{\lambda_j^2} \sum_{kl} V_{ki} V_{li} R_{kj} R_{lj} \\
&= (\sigma^2 + a^2) \sum_i \lambda_i^2 + \beta_w \sum_{ij} \frac{\sigma_i^2(D)}{\lambda_j^2} R_{ij}^2
\end{aligned}
\tag{36}
$$

Thus we have the following constrained optimisation problem

$$
\underset{\boldsymbol{R}, \Lambda}{\text{minimise}} \, \sigma^2 \sum_i \lambda_i^2 + \beta_w \sum_{ij} \frac{\sigma_i^2(D) R_{ij}^2}{\lambda_i^2} \quad \text{s.t. } \lambda_i > 0 \, , \, \boldsymbol{R}^T \boldsymbol{R} = \boldsymbol{I}
\tag{37}
$$

Here we let intuition take over. Taking inspiration from the simplification 1, our ansatz is that $\boldsymbol{R}$ is a permutation matrix and $\lambda_i^4 = \frac{\beta_w \sigma_j^2(D)}{\sigma^2 + a^2}$ (where $i$ and $j$ are related by the permutation). To justify that $\boldsymbol{R}$ should be a permutation matrix in this case, we note that as $\lambda_i^2$ gets smaller to keep $||\boldsymbol{W}||_F^2 = \sum_{ij} \sigma_i^2(D) V_{ij} \frac{1}{\lambda_j^2}$ as small as possible, the $R_{ij}$ must ensure the low valued $\lambda_j^2$ are matched with the low valued $\sigma_i^2(D)$. Intuitively this is done when $\boldsymbol{V}$ is a permutation matrix. Once $\boldsymbol{R}$ is chosen as a permutation matrix, showing that $\lambda_i^4 = \frac{\beta_w \sigma_j^2(D)}{\sigma^2 + a^2}$ is simple - just take derivatives and set to zero. While this is not a full proof, if one derives the KKT conditions, this solution is at least a consistent solution, i.e. it is a minima but may not be the global minima. We note that for specific (small) values of $\beta_w$ it will be the global minima.

As before, our actual aim is to minimise $\mathcal{L}$. Since we are free to choose $\boldsymbol{O}$, and we choose it to make the cross terms in equation 32 go to zero, which is when $\boldsymbol{O}$ has at most one non-zero element per column. This corresponds to disentangled representations.

**No simplifications.** Reminding ourselves of the full objective

$$
\mathcal{L} = (\sigma^2 + a^2) \sum_i \lambda_i^2 + a^2 \sum_{j,k,l \neq k} |F_{jk}||F_{jl}| + \beta_w \sum_{ij} \frac{\sigma_i^2(D)}{\lambda_j^2} \sum_{kl} V_{ki} V_{li} R_{kj} R_{lj}
\tag{38}
$$

In this case $V$ is an arbitrary orthogonal matrix, and $\sigma_i^2(D)$ are the (not necessarily equal) singular values of $D$. One potential ansatz is to assume $R$ is a permutation matrix once again, which then reduces the problem to simplification 2. Again this offers a minima, but not necessarily the global minima.

In sum, we can see that there is always a pressure to disentangle due to the middle term in the loss. However it will have to trade-off against the weight regularization term (last term). Thus we posit that for small values of $\beta_w$ disentanglement is preferred, even for arbitrary $D$.

### A.4 DETAILS OF GENERATED DATASETS

We note that no domain knowledge, like the exact number of factors, is provided too our models at any point. However the dimensionality of the network layer must be at least this number to disentangle all factors.

**Disentanglement in supervised shallow neural networks**.

Starting with 6 independent variables, each sampled from a standard uniform distribution, i.e. $e \in [0,1]^6$. We construct a dataset with $x = Oe$, where $O$ is a random orthogonal matrix (random for each model/seed). The target $y \in \mathbb{R}^6$ is a generated from $y = We$ where $W \in \mathbb{R}^{6 \times 6}$ has elements drawn independently from a standard normal distribution ($W$ different from each model/seed). We generate 200000 data points.

**Disentanglement in supervised deep neural networks**.

This is generated in the same way as above but we additionally set $x \leftarrow x^3$, where the cubing is done element-wise.

**Disentanglement in unsupervised shallow neural networks**.

Starting with 6 independent variables, each sampled from a standard uniform distribution, i.e. $e \in [0,1]^6$. We construct a dataset $y \in \mathbb{R}^{50}$, with $y = We$, where $W \in \mathbb{R}^{50 \times 6}$ has elements drawn independently from a standard normal distribution ($W$ different from each model/seed). We generate 200000 data points.

**Disentanglement in unsupervised deep neural networks**.

Starting with 6 independent variables, each sampled from a standard uniform distribution, i.e. $e \in [0,1]^6$. We construct a dataset $y \in \mathbb{R}^{50}$, with $y = MLP(e)$, where the MLP is a 1-hidden layer network with relu activation function on the hidden layer. The hidden later has dimension 28, and all weights of the network are drawn independently from a standard normal distribution (redrawn from each model/seed). We generate 200000 data points.

## A.5 Network architectures and optimisation details

All code will be released on publication. All hyper-parameters are shown in Table 1. We describe any additional details here. We use the Adam optimiser (Kingma & Ba, 2014). Supervised nets use square error loss. Unsupervised nets use sigmoid cross entropy loss.

**Supervised networks.** $\mathcal{L}_{\text{nonneg}}$, $\mathcal{L}_{\text{activity}}$ and $\mathcal{L}_{\text{weight}}$ are applied to all layers. The shallow Network is a 1-layer MLP with hidden dimensions of [188]. The deep networks are 5-layer MLP with hidden dimensions of [188,186,184,182,180]. We use a squared error loss for $\mathcal{L}_{\text{prediction}}$.

**Autoencoders.** We apply $\mathcal{L}_{\text{nonneg}}$ and $\mathcal{L}_{\text{activity}}$ to the latent layer only. We apply $\mathcal{L}_{\text{weight}}$ to all layers. The shallow autoencoder has no hidden layers in the encoder or decoder. The deep autoencoder has and encoder with hidden dimensions of [500,300,100] and a decoder with hidden dimensions of [100,300,500]. The output has a sigmoid and we use a sigmoid cross entropy loss for $\mathcal{L}_{\text{prediction}}$.

**Variational Autoencoder.** We use an architecture described in Higgins et al. (2017a) (Table 2). We use the standard $\beta$-VAEs loss with the following additions. We apply $\mathcal{L}_{\text{nonneg}}$ to the latent layer only. We apply $\mathcal{L}_{\text{weight}}$ to the dense layers in the decoder only. We *do not* apply $\mathcal{L}_{\text{activity}}$ as an effective norm constrain term already exists in the $\beta$-VAEs loss.

| | **Supervised Net** | | **Autoencoder** | | **VAE** |
|---|---|---|---|---|---|
| | *Shallow* | *Deep* | *Shallow* | *Deep* | *Deep* |
| input dimension | 6 | 6 | 50 | 50 | (64, 64, 3) |
| output dimension | 6 | 6 | n/a | n/a | n/a |
| # parameters | 2450 | 138574 | 1570 | 414870 | 766295 |
| # (hidden) layers | 1 | 5 | enc: 0 
 dec: 0 | enc: 3 
 dec 3 | enc: 6 
 dec 6 |
| latent dimension | n/a | | 10 | | 10 |
| learning rate | 3e-3 | | 1e-4 | | 1e-4 |
| batch size | 128 | | 64 | | 64 |
| # gradient updates | 300000 | | 300000 | | 500000 |
| $\beta_{activity}$ | 1e-3 | | 5e-3 | | n/a |
| $\beta_{weight}$ | 1e-4 | | 1e-3 | | $0.01 \rightarrow 1.0$ |
| $\beta_{nonneg}$ | 2.0 | | 0.5 | | 100.0 |
| $\beta_{VAE}$ | n/a | | n/a | | 1.0 |

Table 1: The values of various hyper-parameters. enc/dec: encoder/decoder. n/a: not applicable.

| **Encoder** | **Decoder** |
|---|---|
| Input: $64 \times 64 \times$ number of channels | Input: 10 |
| $4 \times 4$ conv, 32 ReLU, stride 2 | FC, 256 ReLU |
| $4 \times 4$ conv, 32 ReLU, stride 2 | FC, 4×4×64 ReLU |
| $4 \times 4$ conv, 64 ReLU, stride 2 | $4 \times 4$ upconv, 64 ReLU, stride 2 |
| $4 \times 4$ conv, 64 ReLU, stride 2 | $4 \times 4$ upconv, 32 ReLU, stride 2 |
| FC 256 | $4 \times 4$ upconv, 32 ReLU, stride 2 |
| F2 $2 \times 10$ | $4 \times 4$ upconv, number of channels, stride 2 |

Table 2: Encoder and decoder architecture for the variational autoencoder experiments.

### A.6 SPARSITY DOES NOT PROMOTE DISENTANGLEMENT

Readers may wonder if imposing a sparsity constraint would encourage disentanglement, as ReLUs promote sparsity as well as enforcing nonnegativity. This is not the case, and can be understood intuitively, and also with numerical simulation. For intuition, a sparsity constraint creates a diamond shaped iso-contour in neural space (Fig. 8) left, and so encourages the firing rate distribution to fall inside that diamond, which in the case our our random variables, means maximal entangling. We confirm this intuition in simulation (data and setup the same as Fig. 2) with a variety of sparsity regularization strengths (Fig. 8 right). Thus the reason ReLUs promote nonnegativity is not because they induce sparsity (which they do), but because they enforce nonnegativity.

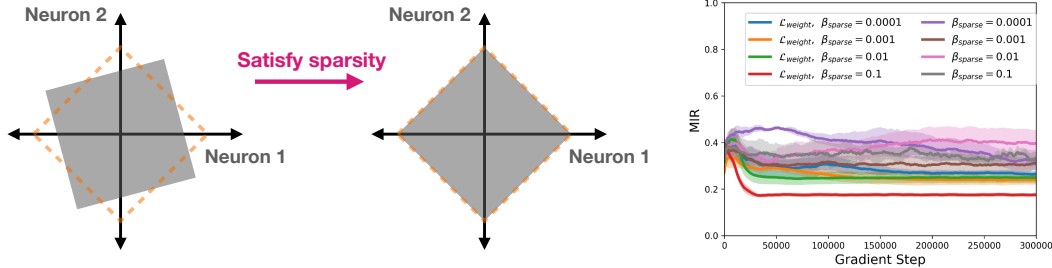

Figure 8: **Sparsity intuition and simulations.** Sparsity does not encourage disentanglement. Here the best situation is maximal entangling. Simulations verify that sparsity constraints do not induce disentanglement.

Similarly, one may wonder if sparsity on weights may encourage disentanglement. We do not observe this empirically (Fig. 9)

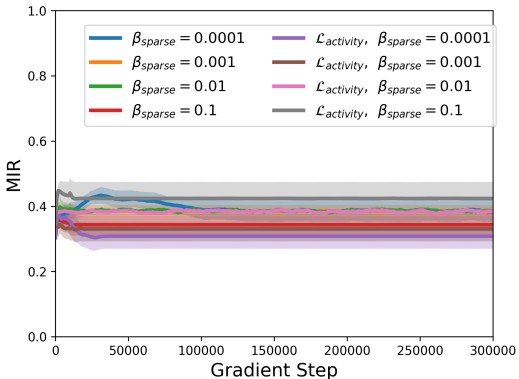

Figure 9: **Sparsity on weights does not lead to disentanglement.**

### A.7 REPRESENTING CATEGORICAL DISTRIBUTIONS

We note that this is a fundamentally different problem to representing factors, since only one category is active at any one time; categories are anti-correlated. This contrasts to factors as all factors can be active independently. So even though the proofs are similar, they are about different situations.

**Theorem.** Given a nonnegative representation, $z \in \mathbb{R}^n$, that linearly reads-out a population, $x \in \mathbb{R}^m$ which is itself a linear projection of a categorical variable (one-hot variable)

$$x = Wz \quad \text{and} \quad x = Cc \tag{39}$$

Where $c \in \mathbb{R}^c$ is a one-hot representation (with each one-hot vector having equal probability) with each element, $c_i$, representing category identity (1 when present, 0 otherwise, only one category active at once), $W \in \mathbb{R}^{m \times n}$ are read-out weights with fixed norm, $||W||_F^2$, and $C \in \mathbb{R}^{m \times c}$ are the data generative weights which is a rank $c$ matrix with all singular values equal and positive; $\sigma_{min}(C) = \sigma_{max}(C) = \sigma(C) > 0$. Then in order to minimise the expected activity energy, $\mathbb{E}||z||^2$, $z$ must be structured so that each neuron only represents at most a single category.

**Intuition.** In order to linearly read-out a categorical variable from a neural population, $z$, then $z$ must take be a linear combination of a one-hot vector ($c$; each element in the one-hot vector corresponding to each category).

$$z = Mc \tag{40}$$

For this representation to be nonnegative, all entries of $M$ must be nonnegative. This means that the population response, $z_i$, for each category, $i$, must be a vector in the positive orthant (Fig. 10). Assuming we keep the expected activity energy constant, i.e.

$$\mathbb{E}||z||^2 = \frac{1}{c}\sum_i ||z_c i||^2 = \frac{1}{c}||M||_F^2 = \text{constant} \tag{41}$$

Then the easiest representation to read-out from is the one-hot representation, since in all other situations the vectors are closer together which require larger read-out weights to disambiguate. Alternatively, if there is noise on $z$ then the categories will be more often confused if the $z_c$ vectors are closer to each other. A one-hot representation is the best, and it is a disentangled representation for each category.

**Proof.** We aim to show when minimising $\mathbb{E}||z||^2$, when $z$ is nonnegative ($z_i \geq 0$), and for a finite norm of the read-out weights, $||W||_F^2$, then it is optimal for each element, $z_i$ in $z$ to represent at most a single category. This is a constrained optimisation problem

$$\underset{W}{\text{minimise}}\, \mathbb{E}||z||^2 \quad \text{s.t. } z_i \geq 0\,,\, Cc = Wz\, \forall c \tag{42}$$

Since we read-out with zero error, $Cc = Wz$, then $W$ must contain all the information of $C$, i.e. $W = CF^+$, where $F^+ \in \mathbb{R}^{c \times n}$ is a rank $c$ matrix. Thus $z$ must be

$$\begin{aligned}
z &= W^+x \\
&= FC^+Cc \\
&= Fc
\end{aligned} \tag{43}$$

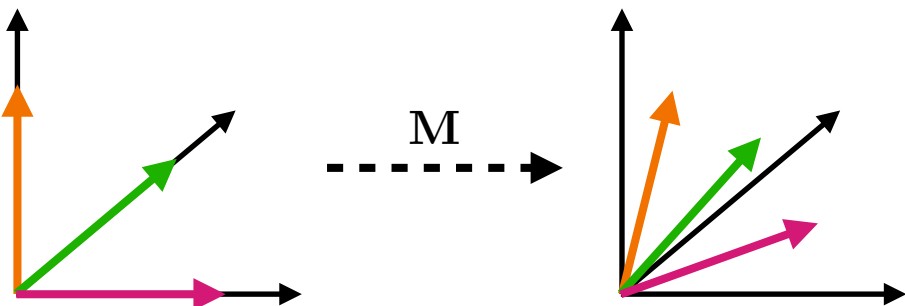

Figure 10: Schematic for representing categories. Each axis is neural firing rate. Each colour is a population activity for a single category, i.e. $z_c$.

For this to be nonnegative, since $c$ is one-hot, then $F_{ij} \geq 0$. Additionally, the expected activity energy becomes

$$
\begin{aligned}
\mathbb{E}||\boldsymbol{z}||^2 &= \mathbb{E}||\boldsymbol{F}\boldsymbol{c}||^2 \\
&= \frac{1}{c}||\boldsymbol{F}||_F^2 \\
&\geq \frac{1}{c}\frac{c^2}{||\boldsymbol{F}^+||_F^2} \\
&= \frac{1}{c}\frac{c^2\sigma^2(\boldsymbol{C})}{||\boldsymbol{W}||_F^2}
\end{aligned}
\tag{44}
$$

Where the final equality if from equation 22, and where the inequality is from equation 17. This inequality becomes an equality if and only if $\boldsymbol{F}^+$ has (scaled) orthonormal rows (or $\boldsymbol{F}$ has (scaled) orthonormal columns). Thus to minimise $\mathbb{E}||\boldsymbol{z}||^2$ we want $\boldsymbol{F}^+$ to have (scaled) orthonormal rows or equally $\boldsymbol{F}$ to have (scaled) orthonormal columns. To satisfy nonnegativity we wanted $F_{ij} \geq 0$. There is only one type of matrix with (scaled) orthonormal columns that has nonnegative entries, and that is when $\boldsymbol{F}$ has rows that contain at most one non-zero element, and so each neuron only 'attends' to one category; categories are disentangled.

### A.7.1 SIMULATION RESULTS

We train an autoencoder on data from 6 categories. The data from each category is noiseless, so we essentially just have 6 data samples that we train on. Each category vector is sampled from a uniform distribution. We train an autoencoder with a deep encoder (hidden layers: [500,300,100]) and a shallow decoder (0-hidden layers), and with latent dimension of 10.

Along with variants of our constraints, we also train a network with a sparsity inducing loss: $\mathcal{L}_{\text{sparsity}} = \beta_{sparsity}\sum_i |a_i|$, where $a_i$ is the activity of a neuron in the latent layer. All other hyper-parameters are as described in the autoencoder section of Table 1, and we set $\beta_{sparsity}$ to be the same as $\beta_{nonneg}$.

We see that only our constraints, and the sparsity inducing loss, achieves disentanglement of categories (Fig. 11).

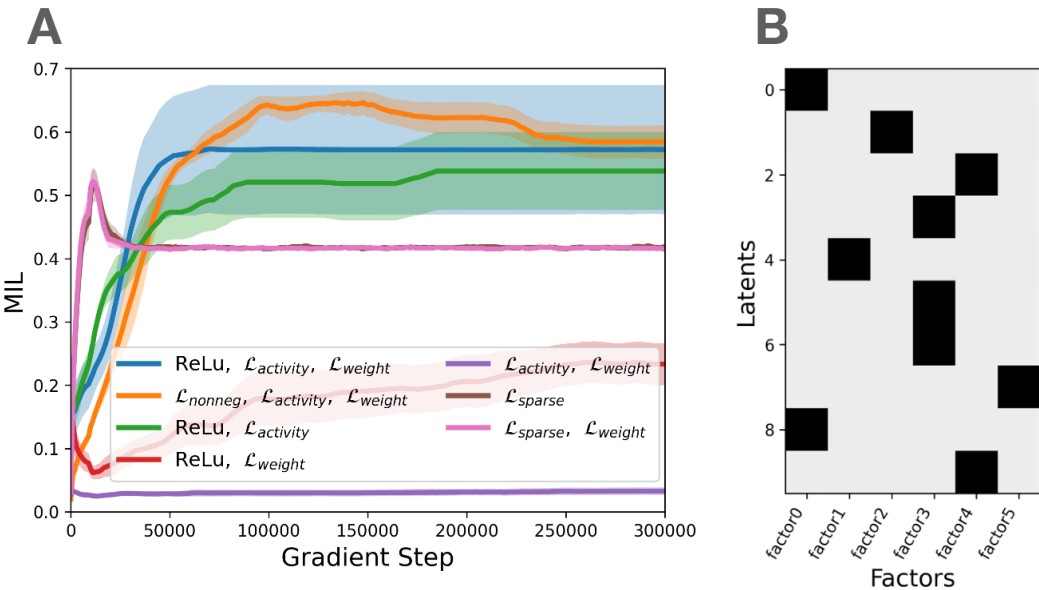

Figure 11: **Learning data generative factors with autoencoders. A)** Training linear autoencoders on linear data. Only models with our constraints, or with sparsity, learn disentangled representations. **B)** Example mutual information matrix from a high MIR model. All learning curves show mean and standard error from 4 mean from 5 random seeds.

## A.8 FURTHER VAE SIMULATIONS RESULTS

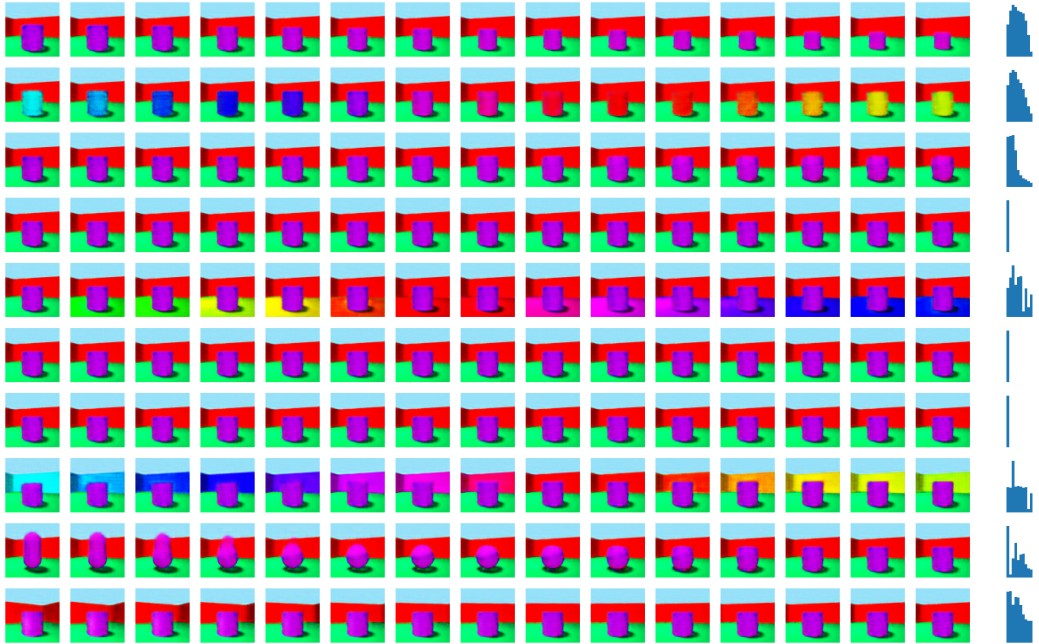

Figure 12: **Latent space traversals.** An image is encoded and then the value of a single latent dimensions is changed, and the resulting image is generated. Each row shows this image for when a single latent dimension is traversed. The histogram shows the marginal distribution of that latent variable. From the images, we can see that each latent dimension predominantly cares about a single ground truth factor.

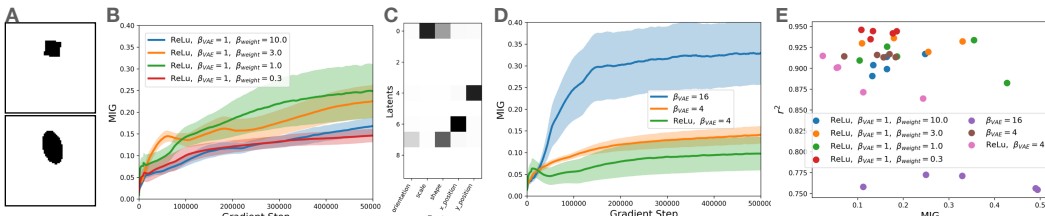

Figure 13: **Same as Fig. 5, but for the dSprites dataset. A)** We train on the dSprites dataset, with two example images shown. These images have 5 underlying factors. **B)** MIG scores are higher with higher weight regularization, and generally higher than any $\beta$-VAE (panel D). **C)** Mutual information matrix for a high scoring model. We note it has dropped a latent dimension. **D)** $\beta$-VAE MIG scores. We note the high disentanglement $\beta$-VAE models have bad reconstruction as they drop latent dimensions. **E)** MIG score against $R^2$ shows models with our constraints lie in the Goldilocks region of high disentanglement and high reconstruction. All learning curves show mean and standard error from 5 random seeds.

### A.9 PATTERN FORMING DETAILS

#### A.9.1 DETAILS ON TRAINING DATA

**A factorised task for rodents.** We generate many 16x16 worlds (so 256 locations; $n_l = 256$) where objects can be at random locations. The number of objects in each world is chosen uniformly between 0 and 8. In practice it is not always possible to place every object as we do not permit objects to be closer than 8 units away.

From each world we have the following data: $\{\boldsymbol{x}, \boldsymbol{o}(\boldsymbol{x}), \boldsymbol{a}(\boldsymbol{x})\}$ for all 256 locations $\boldsymbol{x}$, where $\boldsymbol{x}$ is a one-hot identifier of location, $\boldsymbol{o}(\boldsymbol{x})$ is a binary variable of whether there is an object or not at location $\boldsymbol{x}$, and $\boldsymbol{a}(\boldsymbol{x})$ is a 4-dimensional vector describing the optimal action - a one or zero in each entry corresponding to North/South/East/West (more than one action could be optimal at each location). Optimal actions are towards the closest object.

We train on this data, with a batch size of 8, i.e. presenting 8 such worlds simultaneously. Each world in the batch is randomly sampled from the above process.

**A entangled task for rodents.** This is the same as the above, except we sample only 1 world throughout training, i.e. every world in the 8 batches are identical and all batches are identical.

#### A.9.2 MODEL DETAILS

We use a discrete 16x16 world (so 256 locations; $n_l = 256$) and optimise an independent representation, $\boldsymbol{z}(\boldsymbol{x}) \in \mathbb{R}^{n_c}$, at each location. We now detail each component of the following loss

$$\mathcal{L} = \underbrace{\mathcal{L}_{\text{nonneg}} + \mathcal{L}_{\text{activity}} + \mathcal{L}_{\text{weight}}}_{\text{Biological constraints}} + \underbrace{\mathcal{L}_{\text{location}} + \mathcal{L}_{\text{actions}} + \mathcal{L}_{\text{objects}}}_{\text{Functional constraints}} + \underbrace{\mathcal{L}_{\text{path integration}}}_{\text{Structural constraints}} \tag{45}$$

**Biological losses.** These are exactly the same as in the main paper, but we must also average over all locations, $\boldsymbol{x}$

$$\mathcal{L}_{\text{nonneg}} = \frac{\beta_{\text{nonneg}}}{n_l} \sum_{x} \sum_{i} \max(-z_i(\boldsymbol{x}), 0) \tag{46}$$

$$\mathcal{L}_{\text{weight}} = \beta_{\text{weight}} \sum_{t} ||\boldsymbol{W}_t||^2 \tag{47}$$

$$\mathcal{L}_{\text{activity}} = \frac{\beta_{\text{activity}}}{n_l} \sum_{x} ||\boldsymbol{z}(\boldsymbol{x})||^2 \tag{48}$$

where $z_i(\boldsymbol{x})$ is a neuron in representation $\boldsymbol{z}(\boldsymbol{x})$, $t$ indexes the task (i.e. object, action, location prediction), and the $\beta$ values determines the regularization strength.

**Location loss.** The representation predicts location (a one-hot encoding describing each of the 256 locations) via a linear transformation, which is then fed into a softmax cross-entropy loss. In particular, the logits for each location, $\boldsymbol{x}$, are $\boldsymbol{W}_l \boldsymbol{z}(\boldsymbol{x})$, where $\boldsymbol{W}_l \in \mathbb{R}^{n_l \times n_c}$. If we denote each row of $\boldsymbol{W}_l$ as $\boldsymbol{l}_{\boldsymbol{x}}$, noting that this row that 'corresponds' to location $\boldsymbol{x}$ in the one-hot encoding, then the loss is as follows

$$\mathcal{L}_{\text{location}} = -\frac{\beta_{\text{location}}}{n_l} \sum_{\boldsymbol{x}} \ln \frac{e^{\boldsymbol{l}_{\boldsymbol{x}} \cdot \boldsymbol{z}(\boldsymbol{x})}}{\sum_{\boldsymbol{x}'} e^{\boldsymbol{l}_{\boldsymbol{x}'} \cdot \boldsymbol{z}(\boldsymbol{x})}} \tag{49}$$

**Object loss.** An object is either present or not present at each location, so we use a sigmoid cross-entropy loss. In particular, the logits for each location $\boldsymbol{x}$ is $\boldsymbol{W}_o \boldsymbol{z}(\boldsymbol{x})$, where $\boldsymbol{W}_o \in \mathbb{R}^{1 \times n_c}$. If $\mathbb{1}_{\text{object at } \boldsymbol{x}}$ returns a 1 if an object is present at location, $\boldsymbol{x}$, and 0 otherwise, then the loss is as follows

$$\mathcal{L}_{\text{object}} = -\frac{\beta_{\text{object}}}{n_l} \sum_{\boldsymbol{x}} \mathbb{1}_{\text{object at } \boldsymbol{x}} \ln \sigma(\boldsymbol{W}_o \boldsymbol{z}(\boldsymbol{x})) + \mathbb{1}_{\text{object not at } \boldsymbol{x}} \ln(1 - \sigma(\boldsymbol{W}_o \boldsymbol{z}(\boldsymbol{x}))) \tag{50}$$

**Action loss.** More than one action can be correct at a location, so we use a sigmoid cross-entropy loss for each of the 4 actions. In particular, the logits at location, $\boldsymbol{x}$, are $\boldsymbol{W}_t \cdot \boldsymbol{z}(\boldsymbol{x})$, where $\boldsymbol{W}_t \in \mathbb{R}^{n_a \times n_c}$, where $n_a$ is the number of actions (4 in our case - North, South, East, West). We denote each row

of $\boldsymbol{W}_t$ as $\boldsymbol{t}_j$, noting this row 'corresponds' to action $j$ in the action encoding. If $\mathbb{1}_{a=\boldsymbol{a}(\boldsymbol{x})}$ returns a 1 if action, $a$, is a correct action at location $\boldsymbol{x}$, and 0 otherwise, then the loss is as follows

$$\mathcal{L}_{\text{action}} = -\frac{\beta_{\text{action}}}{n_l} \sum_{\boldsymbol{x}} \sum_{a} \mathbb{1}_{a=\boldsymbol{a}(\boldsymbol{x})} \ln \sigma(\boldsymbol{t}_a \cdot \boldsymbol{z}(\boldsymbol{x})) + (1 - \mathbb{1}_{a=\boldsymbol{a}(\boldsymbol{x})}) \ln(1 - \sigma(\boldsymbol{t}_a \cdot \boldsymbol{z}(\boldsymbol{x}))) \quad (51)$$

**Structural loss.** We use a squared error loss for the structural constraint, which asks for neighbouring representations to be related to each other by an action matrix $\boldsymbol{W}_a$ for each action, $a$. This is just like a path integration loss. This loss is done for every location, $\boldsymbol{x}$, and each of the 4 actions, $a$.

$$\mathcal{L}_{\text{path integration}} = \frac{\beta_{\text{path integration}}}{n_l} \sum_{\boldsymbol{x}} \sum_{a} ||\boldsymbol{z}(i) - f(\boldsymbol{W}_a \boldsymbol{z}(\boldsymbol{x} - \boldsymbol{d}_a)||^2 \quad (52)$$

Where $\boldsymbol{W}_a \in \mathbb{R}^{n_c \times n_c}$ is a weight matrix that depends on action, $a$, (i.e. there are 4 trainable weights matrices - one for each action). $\boldsymbol{d}_a$ means the displacement in the underlying space (the space of $\boldsymbol{x}$), that the action $a$ corresponds to.

This loss is effectively a path integration loss and can be understood by considering the path integration equation. While there are more than one potential forms for path integration we choose the form relating to Gao et al. (2021) which sequentially updates representations in time via:

$$\boldsymbol{z}_t = f(\boldsymbol{W}_a \boldsymbol{z}_{t-1}) \quad (53)$$

If the representation at time-step $t-1$ corresponds to location $\boldsymbol{x} - \boldsymbol{d}_a$, then for path integration to be successful, the representation at time $t$ must correspond to location $\boldsymbol{x}$. Then the equation becomes like equation 3. This is exactly what our loss asks for. We note, however, that this loss on its own could just make all representations at all locations the same constant (and learn all $\boldsymbol{W}_a$ to be the identity), in which case location isn't represented at all. However the location loss ensures all locations are represented differently. So really it is the structural loss and the location loss together that are like path integration. Overall, this formulation translates sequential path integration in time into an optimisation over representation at locations without needing to incorporate time. This greatly reduces the difficulty of optimisation and makes the problem more general than just sequences in time.

**Pattern forming dynamics.** The overall loss can be optimised with respect to the weights. However, it can also be optimised directly with respect to $\boldsymbol{z}$. This is particularly interesting for us, as it allows our representation to be dynamic and change rapidly for a single task, and not just slowly via learning over many tasks. This is a necessity for us as we need to represent objects which may move between tasks. To optimise both $\boldsymbol{z}$ (task particularities) and weights (task generalities), we do so in two stages. First, we optimise with respect to $\boldsymbol{z}$ to 'infer' a representation for the current task. Second, we optimise with respect to the weights to learn parameters that are general across tasks.

When optimising with respect to $\boldsymbol{z}$ we only optimise two terms in the loss: $\mathcal{L}_{\text{objects}}$ and $\mathcal{L}_{\text{path integration}}$. We optimise the first term so the systems has the ability to know where the objects are. We optimise the second term so that information can be propagated around (effectively via path integration).

The dynamics of the $\mathcal{L}_{\text{objects}}$ are:

$$\frac{d\mathcal{L}_{\text{objects}}}{d\boldsymbol{z}(\boldsymbol{x})} = -\boldsymbol{W}_{objects}^T(\mathbb{1}_{\text{object at } \boldsymbol{x}} - \sigma(\boldsymbol{W}_o \boldsymbol{z}(\boldsymbol{x}))) \quad (54)$$

This says if you get the object prediction wrong, then update $\boldsymbol{z}$ to better predict the object. We restrict this update to only take place where the object is, so it is just an object signal. This update is equivalent to a rodent observing that it is at an object.

The dynamics of the $\mathcal{L}_{\text{path integration}}$ are:

$$\begin{aligned}
\frac{d\mathcal{L}_{\text{path integration}}}{d\boldsymbol{z}(\boldsymbol{x})} = \sum_{a} & -(\boldsymbol{z}(\boldsymbol{x}) - f(\boldsymbol{W}_a \boldsymbol{z}(\boldsymbol{x} - \boldsymbol{d}_a)) \\
& + \boldsymbol{W}_a^T (\boldsymbol{z}(\boldsymbol{x} + \boldsymbol{d}_a) - f(\boldsymbol{W}_a \boldsymbol{z}(\boldsymbol{x}))) \odot f'(\boldsymbol{W}_a \boldsymbol{z}(\boldsymbol{x}))
\end{aligned} \quad (55)$$

The two terms in the above equation can be easily understood. The first says that the representation at each location, $\boldsymbol{z}(\boldsymbol{x})$, should be updated according to what its neighbours think it should (this is

the same update rule as path integration!). The second term says the representation at each location, $z(x)$, should be updated if did not predict its neighbours correctly. This equation tells representations to update based their neighbours. This is just like a **cellular automata**, but instead of a discrete value being updated on the basis of its neighbours, it is a whole population vector whose elements can are continuous. Indeed, just like cellular automata, it is also possible to initialise a single 'cell' (location) of the cellular automata, and have that representations propagate throughout the space. In this case, it's just like path integration, but spreading through all space at once. We note, however, that in our simulations we initialise representations at all locations (for each task).

We note that while we simulated this on a discrete grid, the same principles apply to continuous cases. In this case the sums over location/actions need to be replaces with integrals.

This is a very general approach for understanding representations. The structural loss does not have to related to the rules of path integration. It can be anything. It could be the rules of a graph. It could be rules of topology. It could have one set of rules at some locations and another set of rules at other locations. The rules don't have to be neighbouring representations telling each other what to be, it could also be long range rules too. If there are structure or rules in the world or behaviour, our constraints say that representations should understand that structure or rules. In mathematics this is known as a homeomorphism. In sum, understanding representations via constraints is very general.

**Hyper-parameters.** $\beta_{\text{nonneg}}, \beta_{\text{weight}}, \beta_{\text{activity}}, \beta_{\text{location}}, \beta_{\text{object}}, \beta_{\text{action}}, \beta_{\text{path integration}}$ are chosen as $1e-2$, $1e-7, 1e-3, 1e-2, 1e-1, 1e-2, 40$. The learning rate is $12e-4$. In the Euler updates for pattern formation, the step size for path integration is $0.1$, and the step size for objects is $1$. We additionally clip the norm of the update in pattern formation for stability. The batch size is 8, i.e. we present 8 worlds simultaneously.

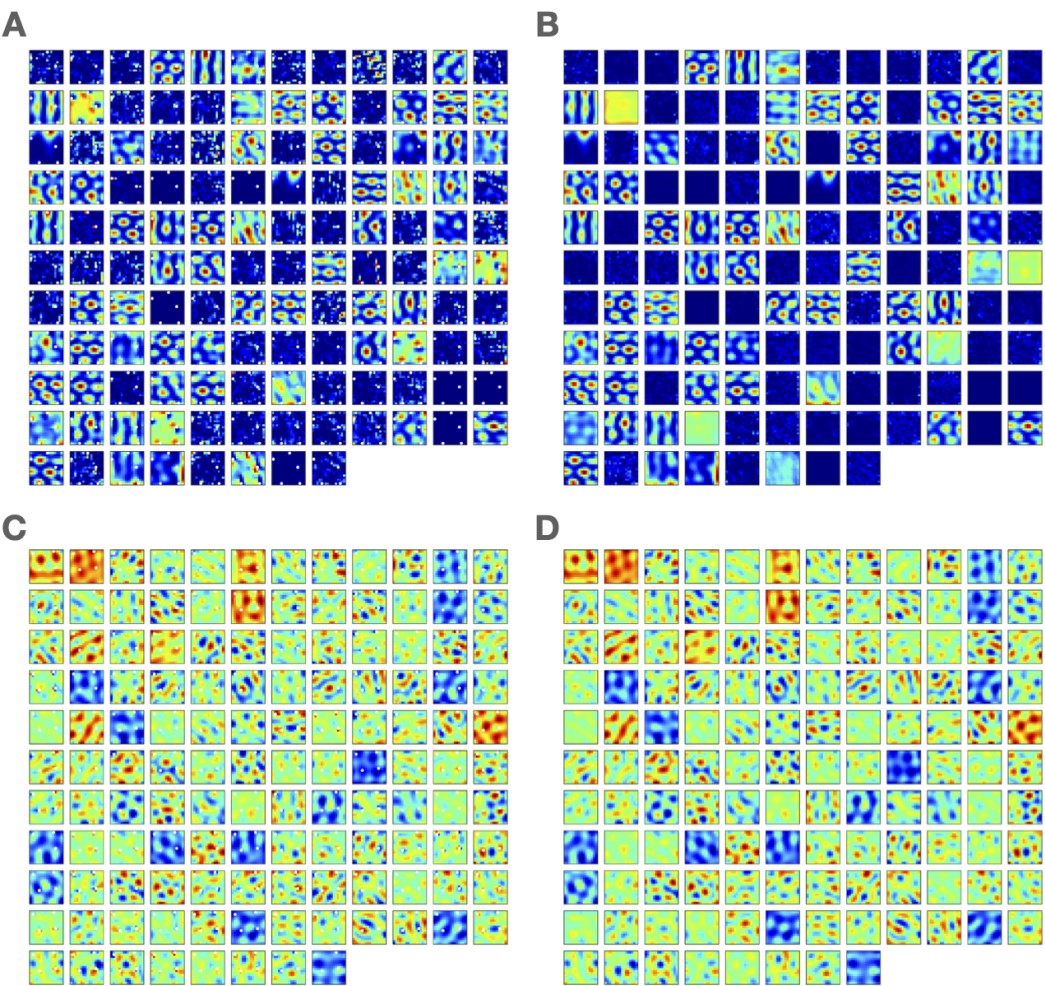

Figure 14: All cells. A/B) Ratemaps from a ReLU model. A) A task with no objects. B) Task with objects. C/D) Ratemaps from a model with linear activation function. C) A task with no objects. D) Task with objects.

