# OpenReview forum: "Disentanglement with Biological Constraints: A Theory of Functional Cell Types"
_ICLR.cc/2023/Conference — ICLR 2023 notable top 25%_

### Official Review · Reviewer_4iCk · 2022-10-24

**Confidence:** 4
**Correctness:** 4
**Technical Novelty And Significance:** 4
**Empirical Novelty And Significance:** 4
**Recommendation:** 10

**Clarity, Quality, Novelty And Reproducibility:**

Beautifully presented paper with adroit insights and contributions to ML and Neuro. The work appears reproducible but there's no code release.

**Strength And Weaknesses:**

Strengths:

- Great introduction.

- Love how the authors presented and explained the proof relating non-neg and efficiency to independent task factors.

- The related work brings together the parallel investigations of disentangled factors and the use of nonnegativity to do so (e.g., Lee & Seung, 2000) in Neuroscience and ML.

- Honestly I could go on and on, but this is a brilliant paper. I love how it begins with a simple concept, explains it like I'm five, and then shows that the theory holds in complex situations that are relevant to the large-scale ML and Neuro pursuits of today.


Weaknesses:

- Can you add a github link with your code?
- Make the text bigger in your figures? It's a little hard to make out. That's it.

Questions:

1. Are the colored circles in Figure 2 cartoons or real data?
2. In Fig 2, if I'm understanding this correctly, relu and a non-neg activity constraint yield the same performance. Does the regularization also affect the signs of the model weights?
3. There's no constraint for non-negative weights, right? Just non-negative activities?
4. What’s the explanation for neurons that show mixed selectivity? Is that just a relaxation on the energy constraint?

**Summary Of The Paper:**

The authors investigate how biological brains could achieve disentangled representations. Through a combination of theory and modeling, they provide ample evidence for two key constraints: (1) nonnegative activities and (2) a constraint on energy expenditure in model/real neurons. The work is exceptionally well explained and builds out from these simple principles to explain results in modeling complex artificial data and rodent experiments.

**Summary Of The Review:**

This paper is an absolute accept. Beautifully written, deeply insightful, great lit review, and lays out theory that I believe can inform the development of better models for AI and the design of new neuro experiments. Best paper I've reviewed for a conference in years (ever?). I'm going to be a jerk and give it an 8 for now just because the authors didn't release their code. Assuming they do over the discussion period I'll bump them to 10.

---

> ### Author Response · Authors · 2022-11-14
> **Response (1/1)**
>
> > The authors investigate how biological brains could achieve disentangled representations. Through a combination of theory and modeling, they provide ample evidence for two key constraints: (1) nonnegative activities and (2) a constraint on energy expenditure in model/real neurons. The work is exceptionally well explained and builds out from these simple principles to explain results in modeling complex artificial data and rodent experiments.
>
> Many thanks!
>
> > Great introduction. Love how the authors presented and explained the proof relating non-neg and efficiency to independent task factors. The related work brings together the parallel investigations of disentangled factors and the use of nonnegativity to do so (e.g., Lee & Seung, 2000) in Neuroscience and ML. Honestly I could go on and on, but this is a brilliant paper. I love how it begins with a simple concept, explains it like I'm five, and then shows that the theory holds in complex situations that are relevant to the large-scale ML and Neuro pursuits of today.
>
> Many thanks!
>
> > Can you add a github link with your code?
>
> We have now included the code as Supplementary Material in this submission.
>
> > Make the text bigger in your figures? It's a little hard to make out. That's it.
>
> We have now increased the size of the text in the figures.
>
> > Are the colored circles in Figure 2 cartoons or real data?
>
> Good spot! They are cartoons - we have now amended the text in the caption.
>
> > In Fig 2, if I'm understanding this correctly, relu and a non-neg activity constraint yield the same performance.
>
> As shown in the figure, they both disentangle approximately the same amount. If however by performance you mean how well they fit the data, then they both match the data virtually exactly, however the ReLu is ever so slightly worse (r^2 0.99995 vs 0.99998, figure not shown here).
>
> > Does the regularization also affect the signs of the model weights? There's no constraint for non-negative weights, right? Just non-negative activities?
>
> The regularisation does not constrain the sign of the weights. We just used l2 regularisation. Weights in neuroscience can have both excitatory and inhibitory effects, and so we have not tried constraining the sign of model weights. We have, however, tried sparsity on the weights but we did not observe any disentanglement (new appendix A.6 figure).
>
> > What’s the explanation for neurons that show mixed selectivity? Is that just a relaxation on the energy constraint?
>
> That’s a really good question. While we definitely do not yet understand all possible reasons for mixed-selectivity, we can name a few. First, when the factors need to be combined nonlinearly, then mixed selectivity is unavoidable. This is necessary all over the brain as the compositions in our world are not always linear. Second, since our theory is one of optimality, mixed selectivity will occur in the learning process towards this optimum. Third, while theory is about composing factors, this is not always the best way to understand situations, and indeed can only understand situations that are composed of those factors. Other general methods such as kernel regression may be more appropriate in those situations, in which mixed selectivity is prominent. We hope the sections ‘When to disentangle?’ and ‘Mixed-selective cells in the brain’ from the Discussion help here. We are happy to amend it if it is not clear.
>
> > Beautifully presented paper with adroit insights and contributions to ML and Neuro. The work appears reproducible but there's no code release.
>
> Many thanks! Code now provided!

---

> > ### Comment · Reviewer_4iCk · 2022-11-17
> > **Response**
> >
> > > Many thanks! Code now provided!
> >
> > Thank you! Bumped my score.

---

> > > ### Author Response · Authors · 2022-11-17
> > > **Many thanks!**
> > >
> > > Many thanks!

---

### Official Review · Reviewer_pBZV · 2022-10-24

**Confidence:** 3
**Correctness:** 4
**Technical Novelty And Significance:** 3
**Empirical Novelty And Significance:** 3
**Recommendation:** 8

**Clarity, Quality, Novelty And Reproducibility:**

Clarity:

The paper is overall well organized and well written.
Section 4, the rodent navigation task, and the problem definition need some revisions and more clarifications, including Eq. 4, Figure 6 and 7.


Quality:

The paper is technical with enough theoretical backing. Theorem 1 shows that in the presence of iid latent factors and nonnegativity of neural activities with constant variance, the representations that achieve minimal activity energy also exhibit disentangling. Theorem 2 generalizes the result of the first theorem by considering generative model for neural representations that can exactly predict the observed data, without requiring constant variance.

Novelty and Originality:

The paper is addressing a challenging and important problem. The introduced neurobiological perspective and neural representation modeling are novel and interesting.

Reproducibility:

I cannot comment on reproducibility. The code is not available.




**Details Of Ethics Concerns:**

no ethics concerns.

**Strength And Weaknesses:**

Strengths:

- The paper is addressing a challenging and important problem, learning disentangled and interpretable latent factors, from a neurobiological perspective, which in my opinion is quite interesting and novel. The authors provide a simple set of constraints which lead to a disentangled representation which are quite valuable and potentially could expand the findings of the previous disentanglement studies.

- The main claim of the manuscript is theoretically and empirically justified.

- The simulation studies illustrate that under iid the proposed metric reveal the disentanglement decrement under entanglement attacks, while the comparable measures fail to present it.

- The authors provide multiple experimental settings to evalute the proposed method.


Limitations:

- Although the paper has attempted to make a bridge between neuroscientific findings and the representation learning field, I think the proposed objective is not well-connected to the other studies in this field, except for beta-VAE. It is not clear how the proposed neurobiological constraints improves the already disentangled representations provided by other approaches like factor VAE, weak supervision method (Locatello, ICML, ICLR 2020), or satisfying local isometry (Horan, NeurIPS 2021 ), all of which do not necessarily rely on decorrelation. It is not clear how these neurobiological constraints will complete those approaches. I think the authors need to elaborate more on this.

- IID Assumptions: I understand that making iid assumption for the latent (task) factors makes the theoretical justification easier, but in practice, especially in the case of neurons, it is not correct to assume that all neurons follow the same distribution.

- Domain knowledge: one of the difficulties of learning a disentangled representation is the absence of domain-specific knowledge, like the number of independent factors. According to the problem definition in Eq. 1, to obtain disentangled factors, the network requires access to the true number of K, as I have understood. If this is correct, is there any way to utilize the proposed approach to learn the true number of latent factors?

- The rodent navigation task is a very interesting experimental study. However, the problem formulation in Eq. 4 is quite unclear and confusing.

- There experimental results for benchmark datasets are very limited.

Additional Comment/Questions:

- Is the rodent navigation dataset a new dataset used in this paper? What are the modalities of the given data (measurements)?
- Can you elaborate more on your findings in Figure 7? How should one compare the results in Figures 6 and 7?

- How would this model perform when K is not matched with the true number of factors? In the case of cell type, how can one interpret the results in terms of the function of each neuron?

- How can the proposed method be used to functionally characterized neurons in the other data modalities, e.g. RNA-seq?

- The authors may want to move some of the supplementary materials, e.g. sequential path integration to the main text. It is quit unclear what Eq. 4 is formulating.


**Summary Of The Paper:**

The manuscript introduced a bio-inspired representation learning that guarantees disentanglement in linear network, under some assumptions. The authors showed two neurobiological constraints i.e., nonnegativity and minimum energy efficiency (with respect to activity and weights), lead to linear factorization, under iid assumption of true factors. Additionally, the authors utilized the proposed biological constraints to interpret cell types which are functionally distinct. The authors used multiple simulated scenarios to demonstrate the performance of the proposed approach for linear and non-linear networks, under supervised and unsupervised fashion. Additionally, they reported their model disentanglement performance in comparison with beta-VAE for a few examples of benchmark datasets including Shape3D, dSprites and a rodent navigation task.

**Update after rebuttal:** I have read the author's rebuttal and the latest revision. The authors have satisfactorily addressed most of questions/concerns and improved the manuscript by providing more details and clarification. After reading the rebuttal and the response to the other reviewers, I'm happy to raise my score to help the paper getting accepted.

**Summary Of The Review:**

I think this is a good submission and the proposed neural representation learning using biological constraints is persuasive. However, I think the authors need to improve section 4 and discuss how these findings are generalizable to the other datasets.

---

> ### Author Response · Authors · 2022-11-14
> **Response (1/2)**
>
> Many thanks for your thorough read of this paper and extremely helpful comments. The paper is much better for them!
>
> > Although the paper has attempted to make a bridge between neuroscientific findings and the representation learning field, I think the proposed objective is not well-connected to the other studies in this field, except for beta-VAE. It is not clear how the proposed neurobiological constraints improves the already disentangled representations provided by other approaches like factor VAE, weak supervision method (Locatello, ICML, ICLR 2020), or satisfying local isometry (Horan, NeurIPS 2021 ), all of which do not necessarily rely on decorrelation. It is not clear how these neurobiological constraints will complete those approaches. I think the authors need to elaborate more on this.
>
> Many thanks for the comment. We have now expanded the literature review (now in Appendix A.2) that hopefully addresses these concerns. More broadly, however, this paper is primarily a neuroscience paper, and so while we have shown the biological constraints to be useful for disentangling, we are not trying to be state-of-the-art nor are we trying to say that they are better or worse than other models - only that biological constraints do lead to disentangling. We understand this is frustrating for the ML community side who would want to see detailed comparisons to other ML methods. We will address this in future work.
>
> > IID Assumptions: I understand that making iid assumption for the latent (task) factors makes the theoretical justification easier, but in practice, especially in the case of neurons, it is not correct to assume that all neurons follow the same distribution.
>
> Many thanks. We appreciate this comment, and agree that neurons are generally not iid. We made the assumption, as you said, to each theoretical justification. We have added in a sentence (in a footnote) clarifying this.
>
> *We understand that task factors, or indeed neurons in the brain, are generally not i.i.d., but we have made this assumption for mathematical convenience.*
>
> > Domain knowledge: one of the difficulties of learning a disentangled representation is the absence of domain-specific knowledge, like the number of independent factors. According to the problem definition in Eq. 1, to obtain disentangled factors, the network requires access to the true number of K, as I have understood. If this is correct, is there any way to utilize the proposed approach to learn the true number of latent factors?
>
> Many thanks. The theorem applies to an arbitrary representation, where it is not necessary to know the number of factors. In the subsequent sections, none of the networks know a priori the number of factors or any domain knowledge. To disentangle all factors, however, it is necessary that the number of neurons in that layer has more neurons than the number of factors. We have added in a sentence clarifying this.
>
> *No domain knowledge, like the exact number of factors, is provided. However the dimensionality of the network layer must be at least this number to disentangle all factors.*
>
> > The rodent navigation task is a very interesting experimental study. However, the problem formulation in Eq. 4 is quite unclear and confusing.
>
> We apologise for the confusing description. We have now rewritten this section and have also included further details in appendix A.9. The relevant main text addition is shown below:
>
> *Formally, our task requires predicting spatial location, x, whether an object is observed, o, and the optimal action, a. If objects move between tasks, then p(x, a, o) = p(x)p(a, o), where o and a are not factored since optimal actions are dependent on objects (see Appendix A.9 for details). Our theory says the representation will have two subspaces - one for allo- centric location (for predicting x) and one for location relative to objects (for predicting o and a) - and that these sub-spaces should be represented in separate neural populations when biological constraints are present.*
>
> > There experimental results for benchmark datasets are very limited.
>
> We understand your frustration here. As before, this work is primarily for explaining neuroscience results, as opposed to getting state-of-the-art results. The results we have shown are there to convince readers that biological constraints really do achieve disentanglement. Thus we feel that comparisons are not necessary here. We will do so in future work delving into the machine learning side of this work. If the reviewer feels strongly enough, we can try to include further comparisons, but cannot promise it will get done in time.

---

> > ### Author Response · Authors · 2022-12-05
> > **Second Response**
> >
> > Hi!
> >
> > As the end of the discussion period is fast approaching, do let us know if you have any further comments or concerns to which we can respond. We'd be interested to hear if our comments and changes to the manuscript helped answer your questions!

---

> ### Author Response · Authors · 2022-11-14
> **Response (2/2)**
>
> > Is the rodent navigation dataset a new dataset used in this paper? What are the modalities of the given data (measurements)?
>
> This is an artificial dataset that we have generated ourselves. We have clarified this, and described the dataset in additional detail, in appendix A.9. The relevant additional sentences are shown here
>
> *We generate many 16x16 worlds (so 256 locations; nl = 256) where objects can be at random locations. The number of objects in each world is chosen uniformly between 0 and 8. In practice it is not always possible to place every object as we do not permit objects to be closer than 8 units away.*
>
> *From each world we have the following data: {x, o(x), a(x)} for all 256 locations x, where x is a one-hot identifier of location, o(x) is a binary variable of whether there is an object or not at location x, and a(x) is a 4-dimensional vector describing the optimal action - a one or zero in each entry corresponding to North/South/East/West (more than one action could be optimal at each location). Optimal actions are towards the closest object.*
>
> *We train on this data, with a batch size of 8, i.e. presenting 8 such worlds simultaneously. Each world in the batch is randomly sampled from the above process.*
>
> > Can you elaborate more on your findings in Figure 7? How should one compare the results in Figures 6 and 7?
>
> Figure 7A shows the same results as Figure 6, i.e. for a model trained on a setting where objects move between tasks, however figure 7B shows representations from a model trained on a single task where the objects are in fixed locations. The key difference is that in the 1st case objects move and so their locations are factored from spatial locations, whereas in the 2nd case objects locations are fixed and so entangled with spatial locations. The representations are qualitatively different too. In 7A you get two types of cells - grid cells that are object independent and object dependent cells that are active around objects (like OVCs). Whereas in 7B all cells look similar - they are grid-like but warp their firing fields around objects. Figure 7C quantifies this warping across many different model initialisations - only the entangled task shows this warping. We hope that this clears up the confusion. We currently keep the text as is, though on your further suggestion, we are happy to change text in the paper to help clarify things.
>
> > How would this model perform when K is not matched with the true number of factors? In the case of cell type, how can one interpret the results in terms of the function of each neuron?
>
> K does not need to be matched with the true number of factors. For example there are only two factors in the cell-type experiments, but we use 100 neurons. Similarly, in the unsupervised tasks, the latent layer had 10 neurons, but there were 6 ground truth factors.
>
> > How can the proposed method be used to functionally characterized neurons in the other data modalities, e.g. RNA-seq?
>
> We are not familiar with RNA-seq, though it seems to be for gene sequencing? We are not sure how these methods could be applied there.
>
> > The authors may want to move some of the supplementary materials, e.g. sequential path integration to the main text. It is quit unclear what Eq. 4 is formulating.
>
> Many thanks for the comments. We would like to move much of the appendix to the main text, but we are heavily space limited. Throughout the manuscript, we point to relevant sections in the appendix that go into more detail on sections in the main text. For example, in the paragraph below equation 4, we point to Appendix A.9 that explains the losses in equation 4.
>
> > The paper is overall well organized and well written. Section 4, the rodent navigation task, and the problem definition need some revisions and more clarifications, including Eq. 4, Figure 6 and 7.
>
> We hope that our edits to section 4 (as detailed above) have helped here! We have changed both Figure 6 and 7 as well as the navigation task section (please see paper).
>
> > The paper is addressing a challenging and important problem. The introduced neurobiological perspective and neural representation modeling are novel and interesting.
>
> Many thanks!
>
> > I cannot comment on reproducibility. The code is not available.
>
> We have now included the code as Supplementary Material in this submission.
>
> > I think this is a good submission and the proposed neural representation learning using biological constraints is persuasive. However, I think the authors need to improve section 4 and discuss how these findings are generalizable to the other datasets.
>
> We hope that our improvements to section 4, as well as our extended literature review and clarification that the primary role of this paper is explaining neuroscience results alleviates the reviewer’s concerns.

---

### Official Review · Reviewer_9LCk · 2022-10-24

**Confidence:** 4
**Correctness:** 3
**Technical Novelty And Significance:** 3
**Empirical Novelty And Significance:** 3
**Recommendation:** 6

**Clarity, Quality, Novelty And Reproducibility:**

Generally, the paper lacks a lot of technical details. It is often not clear what the task is, or what the model is that is being evaluated, or what the evaluation metric is (see below).

Random Points:

Please use the more common term ‘disentanglement’ (rather than ‘disentangling’) in the paper, this would at least align with the ML literature. Ideally, go beyond the buzzwords used in ML and relate to the proper terminology from source recovery and ICA.

The last sentence of the abstract is very confusing and I am not sure what the authors are trying to say.

The second paragraph of the introduction is a collection of random facts; the term disentanglement is not defined; what is the message of this paragraph, please clarify.

3rd paragraph: this (https://arxiv.org/pdf/1206.5538) would be the standard reference for sources of variation; this (https://arxiv.org/pdf/2107.08221.pdf&lt;/p&gt;) challenges the idea that disentangled representations generalise better; What is the evidence for the trade-off mentioned in the last sentence?

1.1. Reference for Oja rule? Also, there is a whole line of research missing here. Barlow already postulated that neurons perform redundancy reduction, i.e., minimising mutual information between their representations, i.e., finding independent components. BTW, that would also take care of the multiple neurons representing the same factor issue seen here.
‘While it has beeen shown’ +’that’

D. in brains: here are other references about disentanglement in brains (https://elifesciences.org/articles/43625.pdf) and (https://www.science.org/doi/full/10.1126/sciadv.aax5979). The second to last sentence needs more explanation.

2: what is ‘neural disentangling’? What are a and C? Why is C the same for all neurons? Why minimise the square of the expectation and not the expectation of the square or the variance? Why write |M_jk||M_jl| and not just state that every row of M has exactly one non-zero entry?

What is the dimensionality of x?

Theorem 2: the zero error assumption seems rather unrealistic?

Fig 2: What is the scale of the MI matrices? The whole plot and scripts are too small; Define MIR; What is the noise ceiling in D?

3: 2nd paragraph it is not clear what the task is or what setting is being studied here. One problem with the proposed measure of disentanglement is that for factors a,b and neural response z: I(z;a) and I(z;b) may be low, but I(z;(a,b)) may be high – For instance, if there is a spiral-shaped function with bases a and b and height z – the same problem of pairwise vs. Multi-information measures exists also for the MIG metric.

Page 4 bottom: Again, what is the setting? What is dim(y)?

Fig3: b) what is the noise ceiling?

Disentangling in Deep Nets: how is the data generated, what is the model? Please provide more details.

Disentangling in unsupervised deep nets: You cite Locatello et al. 2019; So you are aware of the fact that this setting is equivalent to nonlinear ICA and that, theoretically, this should not work. Please comment on this. A similar construction as in the original Hyvarinen & Pajunen, 1999 (nonlinear ICA is not identifiable) result can be extended to cover the non-negative setting here. Are the authors claiming that their theory extends to the nonlinear mixing case? If not, please comment why, empirically, it seems to be working to some extent. Maybe because of (https://openaccess.thecvf.com/content_CVPR_2019/papers/Rolinek_Variational_Autoencoders_Pursue_PCA_Directions_by_Accident_CVPR_2019_paper.pdf)? Which could be broken by (http://proceedings.mlr.press/v139/zietlow21a/zietlow21a.pdf)?

‘VAE loss already’ + ’includes’

If you evaluate your method on the dislib benchmark, then please provide proper comparisons against other methods, all metrics and with a sufficient number of seeds (see Locatello et al, 2019).

Fig5: b) what is beta_weight? What is Goldilocks?

A factorised task for rodents: second to last sentence, I don’t see how this falls out from the theory?

The use of the term path integration is misleading at the beginning of the next paragraph.

Fig6: explain the Real OVCS subplot; C) the grids do not appear to be hexagonal or split into modules of different spatial resolution like real GCs? Explain subplots D; explain the spatial correlation and the subplots in H;

Equation 3 needs much more explantation, what is being shown here?

The sum over loss terms in equation 4 is astonishing, one might wonder whether, with that many degrees of freedom, the model can be forced to produce all sorts of results – put differently, if all these hyperparameters can be adjusted, is it informative if, in the end, a setting is found that leads to the presented results?

‘Some cells change a lot, some cells don’t change […] thus, the representation has two disentangled modules’ – what is the quantitative measure for this claim. Honestly, this is frustrating, these are very interesting questions that seem to be dealt with in a hand waivy manner.

Last paragraph section 4: please clarify the distinction and usage of the terms stereotyped behaviour and the concepts of independent factors used so far.

**Strength And Weaknesses:**

Strengths: The paper touches on many interesting and highly relevant topics, including: the relationship between non-negative latent variable models (e.g. NMF) and source separation (ICA); nonlinear source recovery; and the reason for the existence of distinct cell types. The mathematical proof and the linear source recovery simulations appear to be valid.

Weaknesses: It is rather frustrating to see how all these fascinating research questions are dealt with in a short inconclusive manner. The relation between non-negativity and source separation alone would provide more substance than could easily fit into a ICLR conference paper. Ideally, this would then also include a much more thorough literature research and connection to existing source separation and (non-negative) tensor factorisation (like NMF) methods since the early 90’s, as well as a number of additional control studies (changing the number of factors, their distributions, testing violations of different model assumptions, benchmarking against other methods like sparse coding etc.). The other two subjects nonlinear ICA (disentanglement) and cell type emergence are also extremely interesting and deserve a much more thorough treatment.

**Summary Of The Paper:**

I am very impressed by the authors' thorough replies to all the concerns I raised. I can see that a lot of thought and work has gone into improving the papers' clarity and claims. I am raising my score accordingly.

The authors show that non-negativity of neural responses together with an energy constraint encourage disentangled representations. They provide a mathematical proof for this conjecture in two linear settings: linear neural responses, and linear generative models. They demonstrate that this work on linear data and, apparently, also on nonlinearly mixed data. Finally, the show how these constraints encourage distinct cell types in a model of spatial navigation.

**Summary Of The Review:**

This paper addresses some very interesting questions. They provide an interesting mathematical derivation that could provide a substantial contribution at the intersection of ICA and non-negative factorisation (NMF, sparse coding etc.) literature. However, this would need a much more extensive treatment with more details. I think there are big questions related to the nonlinear setting (see Locatello et al., 2019). The arguments on cell types are interesting. However, the RNN model has a lot of degrees of freedom, the experimental results are unclear and one might wonder if this theory of efficient coding and sources recovery (which needs more links to existing literature) would also apply to, e.g., visual neurons.

I strongly encourage the authors to split this up into multiple papers and do a more thorough scientific investigation of the different component claims: 1) ICA and non-negativity; 2) non-negativity as inductive bias in nonlinear ICA?; 3) cell types and non-negativity leading to separated components. Each of these questions is very important and interesting and it would be great to give them a proper treatment that creates new scientific insights, ideally with higher SNR than the average ML paper.

---

> ### Author Response · Authors · 2022-11-14
> **Response (1/6)**
>
> Many thanks for your thorough read of this paper and providing us many useful references. Your comments have helped us make the paper clearer and better - many thanks once again!
>
> > It is rather frustrating to see how all these fascinating research questions are dealt with in a short inconclusive manner. The relation between non-negativity and source separation alone would provide more substance than could easily fit into a ICLR conference paper. Ideally, this would then also include a much more thorough literature research and connection to existing source separation and (non-negative) tensor factorisation (like NMF) methods since the early 90’s, as well as a number of additional control studies (changing the number of factors, their distributions, testing violations of different model assumptions, benchmarking against other methods like sparse coding etc.). The other two subjects nonlinear ICA (disentanglement) and cell type emergence are also extremely interesting and deserve a much more thorough treatment.
>
> Many thanks for the comments. We agree that there are many more interesting research questions here that could be answered in one paper - we will be expanding on this work in subsequent papers. For now though, we think that a single paper with the current ideas is a good start.
>
> We have included a broader literature review in appendix A.2 (due to space constraints), and have included the control study related to sparse coding - sparsity on weights - in appendix A.6 - this does not lead to disentangling.
>
> > Generally, the paper lacks a lot of technical details. It is often not clear what the task is, or what the model is that is being evaluated, or what the evaluation metric is (see below).
>
> We have made edits according to this and the subsequent comments (details in the subsequent responses) and we hope that our revisions now include all the relevant technical details, including task, model, and evaluation.
>
> > Please use the more common term ‘disentanglement’ (rather than ‘disentangling’) in the paper, this would at least align with the ML literature. Ideally, go beyond the buzzwords used in ML and relate to the proper terminology from source recovery and ICA.
>
> We have changed all uses of ‘disentangling’ to ‘disentanglement’
>
> > The last sentence of the abstract is very confusing and I am not sure what the authors are trying to say.
>
> Apologies, we acknowledge this confusing sentence. It was meant to be getting at the fact that neuroscience has little understanding of how different task structures require different neural representations. If there was, that would go a long way to understanding neural firing in the brain. This work - showing that when a task consists of independent factors, neurons in the brain prefer to tune to individual factors - is one of the first to explicitly show exactly how neurons should look under certain types of task. We have clarified this sentence with the following:
>
> *Overall, this work provides a mathematical understanding of why single neurons in the brain often represent single human-interpretable factors, and steps towards an understanding task structure shapes the structure of brain representation.*
>
> > The second paragraph of the introduction is a collection of random facts; the term disentanglement is not defined; what is the message of this paragraph, please clarify.
>
> Many thanks for the comment. The second paragraph is really for neuroscientists, discussing how neurons in the brain often code for single variables of interest. It is an important paragraph to contextualise this work for neuro-minded people. We appreciate however that we have not defined disentanglement early enough. We have not added in the following sentence in the 3rd paragraph of the introduction:
>
> *We define disentanglement as single neurons responding to single factors of variation (see Appendix A.2 for further details).*
>
> > 3rd paragraph: this (https://arxiv.org/pdf/1206.5538) would be the standard reference for sources of variation;
>
> We have added this citation in the broader literature review (appendix A.2).
>
> > this (https://arxiv.org/pdf/2107.08221.pdf</p>) challenges the idea that disentangled representations generalise better;
>
> Many thanks for this citation - we had not seen it before. We have added it in with the following sentence.
>
> *though some work has challenged the idea that disentangled representations generalise better (Schott et al., 2022).*
>
> > What is the evidence for the trade-off mentioned in the last sentence?
>
> Many apologies - we had mistakenly thought that the result was in Locatello et al., 2019, but now cannot can't find it. We have removed the sentence.

---

> > ### Author Response · Authors · 2022-12-05
> > **Second Response**
> >
> > Hi!
> >
> > As the end of the discussion period is fast approaching, do let us know if you have any further comments or concerns to which we can respond. We'd be interested to hear if our comments and changes to the manuscript helped answer your questions!

---

> ### Author Response · Authors · 2022-11-14
> **Response (2/6)**
>
> > 1.1. Reference for Oja rule? Also, there is a whole line of research missing here. Barlow already postulated that neurons perform redundancy reduction, i.e., minimising mutual information between their representations, i.e., finding independent components. BTW, that would also take care of the multiple neurons representing the same factor issue seen here. ‘While it has beeen shown’ +’that’
>
> We have added the reference to Oja’s rule. We have also referred to Barlow, and have stated that while we do not consider parsimony here, that that is also likely an important constraint in biology. Here are the relevant sentences (in the broader literature review - appendix A.2)
>
> *Barlow first hypothesised that neurons represent independent components via redundancy reduction - minimising mutual information between their representation (Barlow, 1961; 1972)*
>
> *We note that Barlow also considered parsimony as a representational pressure too. While we do not consider it in this paper, we believe it is an important constraint.*
>
> > D. in brains: here are other references about disentanglement in brains (https://elifesciences.org/articles/43625.pdf) and (https://www.science.org/doi/full/10.1126/sciadv.aax5979). The second to last sentence needs more explanation.
>
> Many thanks - these are very relevant citations! We have cited both in a broader literature review in appendix A.2 (in appendix due to space constraints).
>
> > 2: what is ‘neural disentangling’? What are a and C? Why is C the same for all neurons? Why minimise the square of the expectation and not the expectation of the square or the variance? Why write |M_jk||M_jl| and not just state that every row of M has exactly one non-zero entry?
>
> We have now included a definition of disentanglement in the introduction, and referred to further details in the appendix A.1.
>
> *We define disentanglement as single neurons responding to single factors of variation (see Appendix A.2 for further details).*
>
> C is the sum of the variance for each neuron, i.e. the population variance, and so is not the same for all neurons. a is the maximum (and - a is the minimum) value that the underlying i.i.d. variables take. Both these points are already stated in Theorem 1 in the main text. We have additionally stated that |M_jk||M_jl| equivalently means that every row of M has exactly one non-zero entry in the following sentence
>
> *$ | M_{jk}| | M_{jl} | = 0 $ for for $ k \neq l$, a.k.a. each row of M has at most 1 non-zero entry.*
>
> > What is the dimensionality of x?
>
> Apologies that this is not stated in the main text. We have changed this now.
>
> > Theorem 2: the zero error assumption seems rather unrealistic?
>
> We agree that it is unrealistic, but for the purposes of this paper we think it is OK. Our future work will include relaxing this assumption.
>
> > Fig 2: What is the scale of the MI matrices? The whole plot and scripts are too small; Define MIR; What is the noise ceiling in D?
>
> The scale of the first MI matrix ranges from 0 to 0.6, the second goes from 0-2.25 (we now mention this in the figure caption). We have increased the size of the plots and text within plots. We now reference the MIR definition in Figure 2 caption. Since we did not add any additional noise to the sampled factors, the noise ceiling is 1. This further highlights the fact that our metric is not the best of all possible metrics, since it is normalised by the mutual information of all factors, rather than just the gap between the factors with 1st and 2nd highest MI, thus it appears lower even with near perfect disentanglement - as can be seen from the MI matrices. Again, we are not trying to introduce a new gold-standard metric, so we feel this is OK and very much still good enough for the central point of this paper. We have added in a paragraph however to state reasons why our metric is not ideal.
>
> *We do not propose this metric as a gold-standard metric for disentanglement, indeed in some situations such as spiral shaped functions, it could exhibit pathological properties. Furthermore since our metric is normalised by the sum mutual information of all factors, rather than the just the gap between factors with 1st and 2nd highest MI, it appears lower even when with near perfect disentanglement.  However in our situations it works just fine. We use it over other metrics since other metrics penalise situations where more than one neuron codes for a ground-truth factor.*

---

> ### Author Response · Authors · 2022-11-14
> **Response (3/6)**
>
> > 3: 2nd paragraph it is not clear what the task is or what setting is being studied here. One problem with the proposed measure of disentanglement is that for factors a,b and neural response z: I(z;a) and I(z;b) may be low, but I(z;(a,b)) may be high – For instance, if there is a spiral-shaped function with bases a and b and height z – the same problem of pairwise vs. Multi-information measures exists also for the MIG metric.
>
> Many thanks. We do not propose this metric as a gold-standard metric for disentanglement. We use it since other metrics penalise situations where more than one neuron codes for a ground-truth factor. While we take your point that the metric is not ideal, for the ground truth factors in our simulations, the metric works fine.. Again our purpose is not trying to claim that our metric is amazing, nor are we in the game of getting state-of-the-art results, this is more a paper that explains neuroscience results. Nevertheless, we take your point and so include a paragraph on this (same as the paragraph shown above).
>
> > Page 4 bottom: Again, what is the setting? What is dim(y)?
>
> We have included the dimension of y in the text (it is 6 for the supervised tasks and 50 for the unsupervised tasks). We have also included further details of all the generated datasets in appendix A.4.
>
> > Fig3: b) what is the noise ceiling?
>
> See above response!
>
> > Disentangling in Deep Nets: how is the data generated, what is the model? Please provide more details.
>
> We have included a detailed description of the data generative methods in the appendix A.4, and the model is described in appendix A.5 (referenced to in the main text).
>
> > Disentangling in unsupervised deep nets: You cite Locatello et al. 2019; So you are aware of the fact that this setting is equivalent to nonlinear ICA and that, theoretically, this should not work. Please comment on this. A similar construction as in the original Hyvarinen & Pajunen, 1999 (nonlinear ICA is not identifiable) result can be extended to cover the non-negative setting here. Are the authors claiming that their theory extends to the nonlinear mixing case? If not, please comment why, empirically, it seems to be working to some extent. Maybe because of (https://openaccess.thecvf.com/content_CVPR_2019/papers/Rolinek_Variational_Autoencoders_Pursue_PCA_Directions_by_Accident_CVPR_2019_paper.pdf)? Which could be broken by (http://proceedings.mlr.press/v139/zietlow21a/zietlow21a.pdf)?
>
> We have now included an expanded literature review in appendix A.2 (due to space constraints) that includes all these citations, and relates this work to ours where possible.
>
> > ‘VAE loss already’ + ’includes’
>
> Good spot! We have amended that sentence.
>
> > If you evaluate your method on the dislib benchmark, then please provide proper comparisons against other methods, all metrics and with a sufficient number of seeds (see Locatello et al, 2019).
>
> We understand your request here. We however want to convey that the point of this paper is not about state-of-the-art disentanglement, or comparisons to other methods, as it is primarily a paper to explain neuroscience results. Thus we feel that comparisons are not necessary here. We will do so in future work delving into the machine learning side of this work. If the reviewer feels strongly, we can try to include further comparisons, but cannot promise it will get done in time (we have spent this rebuttal period so far responding to all other review points).
>
> > Fig5: b) what is beta_weight? What is Goldilocks?
>
> Beta_weight is the regularisation strength of the weight regularisation. It is spoken about in the ‘regularizers as constraints’ paragraph, but we have now included it in the figure caption too.  Apologies ‘Goldilocks’ is a common term originating from an old fairy tale meaning not ‘too much of one thing, not too much of the other’ i.e. the ‘just right’ region. So in this case it means the disentanglement pressure of non-negativity is not too much that it affects performance.

---

> ### Author Response · Authors · 2022-11-14
> **Response (4/6)**
>
> > A factorised task for rodents: second to last sentence, I don’t see how this falls out from the theory?
>
> Many thanks for this - it has made us make things clearer! We have now described the task in terms of variables x, o and a where (space, objects/rewards, and actions respectively). Since the objects/rewards move around between environments p(x,o,a) = p(s)p(o,a) - space and {objects/rewards and actions} are factored, but actions are towards objects/rewards and so are not factoid from each other. Thus if we have a representation that linearly reads out space,  objects/rewards and action, then, from the theory, we expect the space and {object/reward and action} to be represented with different sets of neurons. The new text reads:
>
> *Formally, our task requires predicting spatial location, x, whether an object is observed, o, and the optimal action, a. If objects move between task, then p(x, a, o) = p(x)p(a, o), where o and a are not factored since optimal actions are dependent on objects (see Appendix A.9 for details). Our theory says the representation will have two subspaces - one for allo- centric location (for predicting x) and one for location relative to objects (for predicting o and a) - and that these sub-spaces should be represented in separate neural populations when biological constraints are present.*
>
> > The use of the term path integration is misleading at the beginning of the next paragraph.
>
> We are not sure what the reviewer finds misleading about it. Throughout that paragraph and the following one we have been clear that we have translated the problem of path integrating in a RNN to directly estimating the neural representations of space that afford path-integration. We are happy to make changes, but are unsure what the reviewer has found misleading.
>
> > Fig6: explain the Real OVCS subplot; C) the grids do not appear to be hexagonal or split into modules of different spatial resolution like real GCs? Explain subplots D; explain the spatial correlation and the subplots in H;
>
> We have explained the Real OVC subplot with:
>
> *OVCs encode relative location to objects with firing fields at specific distances and orientations from (white) objects.*
>
> The grid cells are hexagonal. We have now included a grid-score analysis (a measure of being grid cell; Barry et al. 2012) in Figure 6 which shows high scores (above 0.3-0.5 is generally considered a grid cell, and our results show many cells in the range 0.5-1.5). We agree that we only see a single grid module, but the purpose of our work is not to explain every detail of grid cells, but rather explain the distinction between grid cells and object vector cells when tasks are factored.
>
> We have added in an explanation of subplot D with the following text in Figure 6 caption:
>
> *Model ratemaps when trained with a ReLU activation function. Task 1 contains no objects, while task 2 has several objects (white dots). We see two types of cell representation: D) GCs that do not change across tasks and E) OVCs that only appear in the presence of objects.*
>
>  We have explained the spatial correlation subplots with the following text in Figure 6 caption:
>
> *To quantify modules in the ReLu model, we compute the distribution of individual cell spatial correlations across different tasks. H) The mode with high spatial correlation are cells that have high grid-score (Barry et al., 2012) and are grid cells (GCs), the mode with low spatial correlation are cells that respond similarly around objects (OVCs). I) Only one mode of cells is seen without a ReLu activation.*
>
> We have also changed our explanation in the main text (see comment below on ‘Some cells change a lot, some cells don’t change’).
> We have changed the explanation of subplot H to
>
> *To quantify module-ness over many random seeds, we compute the cosine distance between the population’s contribution to x and o. This is done by taking the absolute value of weights projecting from z to x and o, then summing over the space/object dimension (to obtain a vector the same dimension as z), then computing the cosine distance. We see low cosine distance for the ReLu indicating different cells code for space vs objects - i.e. modules.*

---

> ### Author Response · Authors · 2022-11-14
> **Response (5/6)**
>
> > Equation 3 needs much more explantation, what is being shown here?
>
> Many thanks for this - it has helped us clarify things. Path integration is usually done through time by an equation like $z_t = f( W_{a} z_{t-1} ) $. If the representation at time-step $ t-1$ corresponds to location $ x - d_a $, then for path integration to be successful, the representation at time $ t $ must correspond to location $ x $. Then the equation becomes like equation 3. This is exactly what our loss asks for (see appendix A.9). It’s true that this is not exactly like path integration since all representations could just be the identity. But when combined with the location loss (see appendix A.9) locations have to be represented differently. These two losses together are then equivalent to path integration. Due to space constraints, we have added in further explanation in appendix A.9 (under ‘structural loss’). We have referenced this in the main text after equation 3. The text in the appendix is:
>
> *This loss is effectively a path integration loss and  can be understood by considering the path integration equation. While there are more than one potential forms for path integration we choose the form relating to Gao et al. 2021 which sequentially updates representations in time via:
> $z_t = f( W_{a} z_{t-1} ) $
> If the representation at time-step $ t-1$ corresponds to location $ x - d_a $, then for path integration to be successful, the representation at time $ t $ must correspond to location $ x $. Then the equation becomes like equation 3. This is exactly what our loss asks for. We note, however, that this loss on its own could just make all representations at all locations the same constant (and learn all $ W_{a} $ to be the identity), in which case location isn't represented at all. However the location loss ensures all locations are represented differently. So really it is the structural loss and the location loss together that are like path integration. Overall, this formulation translates sequential path integration in time into an optimisation over representation at locations without needing to incorporate time. This greatly reduces the difficulty of optimisation and makes the problem more general than just sequences in time.*
>
> >The sum over loss terms in equation 4 is astonishing, one might wonder whether, with that many degrees of freedom, the model can be forced to produce all sorts of results – put differently, if all these hyperparameters can be adjusted, is it informative if, in the end, a setting is found that leads to the presented results?
>
> The only additional losses, as compared to the earlier ML examples are the three functional losses - these however are required to solve the task - and the structural loss which is required to learn representations that are structured according to space. There isn’t much we can do about that. We agree though that there are lots of moving parts, but still think it is informative being that we have an analytic theory and other simulations that all agree with each other. We do not present a hyperparameter search here because we decided to spend time so far in the rebuttal period responding to other points. We do understand the reviewer’s comments however, and so if the reviewer feels strongly about this, then we will try to do the simulations but cannot promise they will be ready before the paper editing period of rebuttals.
>
> >‘Some cells change a lot, some cells don’t change […] thus, the representation has two disentangled modules’ – what is the quantitative measure for this claim. Honestly, this is frustrating, these are very interesting questions that seem to be dealt with in a hand waivy manner.
>
> Apologies. We have now added a quantitative measure and have shown a double-dissociation between the cells that ‘change a lot’ and the ‘cells that don't’. In particular the cells with representations that do not change between tasks have high grid-score (a measure to determine grid cells) and do not have consistent activity around objects - thus they are grid cells (as can be seen from their rate maps). Whereas cells with representations that change a lot over tasks, have low grid score but instead have consistent firing fields around objects (like object vector cells). These results are now in Figure 6, with a completely reworded caption, and the main text is changed to the following:
>
> *To quantify whether a population really has two distinct cell types (two modules) we analyse the consistency
> of a cell’s representation by taking its average spatial correlation between many different object configurations (Fig. 6G-I). In the ReLu case, there are two modes (Fig. 6G-I) showing a double dissociation: the mode containing cells that don’t change have high grid-score (Barry et al., 2012) and do not have consistent activity around objects. These are GCs. Whereas the mode containing cells that change between tasks have low grid-score and respond consistently around objects (Fig. 6H).*

---

> ### Author Response · Authors · 2022-11-14
> **Response (6/6)**
>
> > Last paragraph section 4: please clarify the distinction and usage of the terms stereotyped behaviour and the concepts of independent factors used so far.
>
> Apologies. In the paper with warped grid cells (Boccara et al., 2019), after the animal finds the 4 rewards, the animal no longer freely explores the environment, but rather sequentially runs between rewards. It does this over and over again, collecting the 4 rewards in sequence using the same spatial trajectories - i.e. a ‘stereotyped behaviour’. We have removed this mention of stereotypes behaviour but made the mention in the previous paragraph clearer with the following sentences:
>
> *Intriguingly, in the warping situation, the rodents exhibited stereotyped behaviour; sequentially running between rewards using the same spatial trajectories rather than freely exploring the space (i.e. behaviour is entangled with space)*
>
> We hope our responses to previous comments (and our newly included text) have made clearer what we mean by independent factors.
>
> > This paper addresses some very interesting questions. They provide an interesting mathematical derivation that could provide a substantial contribution at the intersection of ICA and non-negative factorisation (NMF, sparse coding etc.) literature. However, this would need a much more extensive treatment with more details. I think there are big questions related to the nonlinear setting (see Locatello et al., 2019). The arguments on cell types are interesting. However, the RNN model has a lot of degrees of freedom, the experimental results are unclear and one might wonder if this theory of efficient coding and sources recovery (which needs more links to existing literature) would also apply to, e.g., visual neurons.
>
> We hope that our above responses alleviate your concerns here. We thank you for your enthusiasm on our work though.
>
> > I strongly encourage the authors to split this up into multiple papers and do a more thorough scientific investigation of the different component claims: 1) ICA and non-negativity; 2) non-negativity as inductive bias in nonlinear ICA?; 3) cell types and non-negativity leading to separated components. Each of these questions is very important and interesting and it would be great to give them a proper treatment that creates new scientific insights, ideally with higher SNR than the average ML paper.
>
> This paper will be followed up with both a machine learning paper focusing on non-linear ICA and non-negativity, and a neuroscience paper with data analysis and further explanations of cell types and non-negativity leading to separated components. We do however think that this paper is interesting and worthy of publication in of itself. We hope that we have convinced you of this too!

---

### Official Review · Reviewer_GUiS · 2022-10-26

**Confidence:** 4
**Correctness:** 4
**Technical Novelty And Significance:** 4
**Empirical Novelty And Significance:** 4
**Recommendation:** 8

**Clarity, Quality, Novelty And Reproducibility:**

I found this work to be of very high-quality and consider this to be a novel and original contribution to the field.

**Strength And Weaknesses:**

Strengths:
- I think the paper is exceptionally written and conveys the ideas very well.
- The paper is a nice mix of theory and experiments along with a biological case simulation. The authors also present an intuition accompanying the theorem which nicely illustrates the logic.
- Instead of simply comparing representations at the population level which has been the dominant approach in neuroAI, the paper compares representations at the level of axes, i.e. why the brain and ANNs often develop privileged axes. This is an intriguing question that has received little attention in the field so far and I laud the authors for tackling this with clarity.

Weaknesses:
- I think the justification behind non-negativity (of firing rates) as a biological constraint needs some further discussion in the paper. It is known that the firing activity of single neurons can be suppressed (from baseline rates) and activity is always going to be measured with respect to some reference signal/rate. The constraints the authors frame as ‘biological constraints’ are both routinely employed in modern CNNs through ReLU and l1 regularization on activity/weights. In that context, I found the use of the phrase ‘biological constraints’ to be a bit overblown.
- The authors claim their results provide a potential explanation for concept cells (i.e. grandmother cells) but if energy considerations favor disentangled codes, we would expect to find selective codes for every single concept. Clearly, allowing specialized cells for all concepts in complex domains would be impossible as we’d run out of neural hardware. It would be useful if the authors could discuss the trade-offs or speculate about how such trade-offs might be handled by the brain.
- I understand that space might be a limiting factor but it would be helpful to situate their work within the context of a broader literature on disentangling in machines (besides beta-VAE). There is plenty evidence of disentangling in machines as an emergent phenomenon (not baked in) for e.g. the work of David Bau et al. who found interpretable neurons in vision models, multimodal neurons in CLIP (https://distill.pub/2021/multimodal-neurons/), other work by on superposition/ privileged axes (https://transformer-circuits.pub/2022/toy_model/index.html) that also discusses the role of non-negativity constraints etc. There is also other work on understanding specialization in DNNs that could be relevant: https://openreview.net/forum?id=HreeeJvkue9.


**Summary Of The Paper:**

This paper asks the question of why neural systems often develop a factorized or disentangled representation, i.e. a representation which aligns its variables/latent units with a meaningful factorization of the underlying problem structure. The authors use a combination of theory and simulations to show that biological constraints of non-negative and energy-efficiency together gives rise to disentangled representations. Finally taking the specialized cells in hippocampus like grid cells and object vector cells as a biological case in point, the authors simulate spatial processing in ANNs and demonstrate that biological constraints lead to disentangled codes for space and objects when these factors are independent, but the same constraints lead to entangled codes when these factors do not vary independently in a task, providing a potential explanation (specifically) for why grid cells warp their receptive fields and (more generally) for when and why disentangled codes arise in the brain.

**Summary Of The Review:**

I think the question of why neural systems have a factorised code has received little attention from a computational perspective and this paper is an important step towards filling this gap. For the AI community, understanding principles that promote disentanglement is important as they make generalization/transfer to new tasks easier. This paper illustrates important principles for achieving disentanglement in models (although they are already often implements in ANNs albeit without a true understanding of its consequences).  Overall, I think this is a very interesting paper that makes a novel contribution and would appeal to both the neuroscience and AI community.

---

> ### Author Response · Authors · 2022-11-14
> **Response (1/2)**
>
> Many thanks for your thorough read of this paper and your enthusiasm for our work. Your comments have helped us make the paper clearer and better - many thanks once again!
>
> > I think the paper is exceptionally written and conveys the ideas very well.
> The paper is a nice mix of theory and experiments along with a biological case simulation. The authors also present an intuition accompanying the theorem which nicely illustrates the logic.
> Instead of simply comparing representations at the population level which has been the dominant approach in neuroAI, the paper compares representations at the level of axes, i.e. why the brain and ANNs often develop privileged axes. This is an intriguing question that has received little attention in the field so far and I laud the authors for tackling this with clarity.
>
> Many thanks!
>
> > I think the justification behind non-negativity (of firing rates) as a biological constraint needs some further discussion in the paper. It is known that the firing activity of single neurons can be suppressed (from baseline rates) and activity is always going to be measured with respect to some reference signal/rate.
>
> Many thanks for the comment - it has helped us clarify the paper. We agree that firing below baseline can be viewed as ‘negative’ firing - in fact this is exactly how our proof works! Activity below baseline corresponds to negative/positive values in the distribution, and activity above baseline corresponds to positive/negative values in the distribution. We only require that firing below baseline does not go below zero activity. We have clarified this in the paper with:
>
> *We note that this is consistent with neurons having a baseline firing rate where now activity below baseline corresponds to negative/positive values in the distribution, and activity above baseline corresponds to positive/negative values in the distribution.*
>
> > The constraints the authors frame as ‘biological constraints’ are both routinely employed in modern CNNs through ReLU and l1 regularization on activity/weights. In that context, I found the use of the phrase ‘biological constraints’ to be a bit overblown.
>
> Many thanks for the comment. We don’t claim that these constraints are anything new to the community as they, or similar, have been used many times in the literature. However, putting them under the umbrella of biological constraints and then understanding the impact that biology then has on representations is new. We clarify this in the paper.
>
> *While these constraints of nonnegativity and energy efficiency are not new to the machine learning or neuroscience literature, we package them together to learn new insights into how biological constraints shape the space of neural representation.*

---

> > ### Author Response · Authors · 2022-12-05
> > **Second Response**
> >
> > Hi!
> >
> > As the end of the discussion period is fast approaching, do let us know if you have any further comments or concerns to which we can respond. We'd be interested to hear if our comments and changes to the manuscript helped answer your questions!

---

> ### Author Response · Authors · 2022-11-14
> **Response (2/2)**
>
> > The authors claim their results provide a potential explanation for concept cells (i.e. grandmother cells) but if energy considerations favor disentangled codes, we would expect to find selective codes for every single concept. Clearly, allowing specialized cells for all concepts in complex domains would be impossible as we’d run out of neural hardware. It would be useful if the authors could discuss the trade-offs or speculate about how such trade-offs might be handled by the brain.
>
> Thanks for the comment! To decode N concepts linearly, then you need at least N neurons to do so. All we are saying is that each of those N neurons will favour coding for just one concept, rather than a mixture. It is true that with non-linear decoding you can get away with fewer neurons, but at some point you’ll have to decode into N neurons anyway - and if using neural networks that requires at least N neurons. Our belief is that we don’t require a linear readout of every concept, i.e. some concepts are implicit and not explicitly read out, thus you don’t need a neuron for every concept. We now include the following sentence.
>
> *We note that while our theory says you need at least one neuron per concept that is read-out (true even without disentanglement), it does not say you need a neuron for every possible concept - only for ones that are explicitly read-out.*
>
> > I understand that space might be a limiting factor but it would be helpful to situate their work within the context of a broader literature on disentangling in machines (besides beta-VAE). There is plenty evidence of disentangling in machines as an emergent phenomenon (not baked in) for e.g. the work of David Bau et al. who found interpretable neurons in vision models, multimodal neurons in CLIP (https://distill.pub/2021/multimodal-neurons/), other work by on superposition/ privileged axes (https://transformer-circuits.pub/2022/toy_model/index.html) that also discusses the role of non-negativity constraints etc. There is also other work on understanding specialization in DNNs that could be relevant: https://openreview.net/forum?id=HreeeJvkue9.
>
> Many thanks for the comment and pointing us to those works. We have included these works in a broader literature review in appendix A.2 (due to space constraints).
>
> > I found this work to be of very high-quality and consider this to be a novel and original contribution to the field.
>
> Many thanks!

---

### Author Response · Authors · 2022-11-14
**Summary of our responses!**

Many thanks for all of the comments. They have greatly helped us improve this paper. We have tried to address them all, and we hope you agree that the paper is now greatly improved. We have structured our response as follows:

> The reviewer comments are quoted

Our response is in plain text

*New text (where it isn’t too long!) is in italics.*

 In the new pdf we have highlighted new text and edits in purple for ease of reviewing.

**The main changes to the paper are:**

1. The literature review and related work has been greatly expanded and moved to Appendix A.2 due to lack of space in the main text
2. The description of the rodent task has been rewritten
3. Figure 6 has new analyses and the caption completely rewritten
4. The main text discussion of the rodent task has been rewritten
5. In Appendix A.4 we detail the generated datasets for the networks
6. In Appendix A.6 we include an additional simulations of sparsity on weights
7. In Appendix A.9 we provide further details of the rodent task, and its training.

---

### Comment · Program_Chairs · 2023-03-21
**Congratulations on winning the ICLR 2023 Outstanding Paper Honorable Mention!!**

Hello authors,

Big congratulations! For more details about the award process this year, please see https://blog.iclr.cc/2023/03/21/announcing-the-iclr-2023-outstanding-paper-award-recipients/

Best,

Mengdi, on behalf of ICLR 2023 PCs

---

### Decision · Program_Chairs · 2023-01-20

**Decision:**

Accept: notable-top-25%

**Justification For Why Not Higher Score:**

NA

**Justification For Why Not Lower Score:**

The paper tackles the timely topic of disentanglement. It provides an insightful treatment of the subject combining insighftful theoretical and experimental results. There is very strong support from 2 reviewers and none of the complains raise by the 2 less enthusiastic reviewers appear to be relatively minor in the grander scheme of things.

**Metareview: Summary, Strengths And Weaknesses:**

The authors investigate how disentangled representations can be learned by living brains. The paper provides a combination of theorical and modeling results to show that two key constraints are key: (i) nonnegative neural activity and (ii) a constraint on energy expenditure in model/real neurons. Overall, the reviewers are unanimous in that the paper is very well written and deeply insightful, and that all experiments are carried out well.

**Note From Pc:**

if the above contains the word "oral" or "spotlight" please see: "oral" presentation means -> notable-top-5% and "spotlight" means -> notable-top-25%. As stated in our emails, we are disassociating presentation type from AC recommendations